



# Efficacy of high-resolution satellite observations in inverse modeling of carbon monoxide emissions using TM5-4dvar (r1258)

Johann Rasmus Nüß[1], Nikos Daskalakis[1], Fabian Günther Piwowarczyk[1], Angelos Gkouvousis[2,3], Oliver Schneising[1], Michael Buchwitz[1], Maria Kanakidou[1,2,3], Maarten C. Krol[4,5], and Mihalis Vrekoussis[1,6,7]

[1]Institute of Environmental Physics (IUP-UB), University of Bremen, Bremen, Germany
[2]Environmental Chemical Processes Laboratory (ECPL), Department of Chemistry, University of Crete, Heraklion, Greece
[3]Center for the Study of Air Quality and Climate Change (C-STACC), Institute of Chemical Engineering Sciences, Foundation for Research and Technology Hellas, Patras, Greece
[4]Meteorology and Air Quality, Wageningen University and Research, Wageningen, the Netherlands
[5]Institute for Marine and Atmospheric Research, Utrecht University, Utrecht, the Netherlands
[6]Center of Marine Environmental Science (MARUM), University of Bremen, Germany
[7]Climate and Atmosphere Research Center (CARE-C), The Cyprus Institute, Nicosia, Cyprus

**Correspondence:** Johann Rasmus Nüß (rasmus.nuess@uni-bremen.de), Nikos Daskalakis (daskalakis@uni-bremen.de), and Mihalis Vrekoussis (mvrekous@uni-bremen.de)

**Abstract.** Carbon monoxide in the atmosphere adversely affects air quality and climate, making knowledge about its sources crucial. However, current global bottom-up emission estimates retain significant uncertainties. In this study, we attempt to reduce these uncertainties by optimizing emission estimates for the second half of the year 2018 on a global scale with a focus on the northern hemisphere through the top-down approach of inverse modeling. Specifically, we introduce observations from

the TROPOspheric Monitoring Instrument (TROPOMI) into the TM5-4DVAR model. The emissions are further constrained using NOAA surface flask measurements. We conducted six experiments to investigate the impact of data use in our inversions, varying the a priori emissions and observational datasets.

Notably, the inversion driven by satellite observations alone captures flask measurements south of $55°$ N almost as good as the inversions that included those measurements. This indicates that our method could be suitable for near real-time inversions

based purely on satellite observations. Compared to the bottom-up estimates, all experiments result in strong (by up to $75\%$) broad-scale emission reductions in China and India. In part, the reduction over China can be attributed to policy changes. Additionally, the OH climatology used to simulate chemical loss appears to be underestimated in that region, which also skews the inversions towards lower emissions. Conversely, in most experiments, we find strong localized emission increments over Europe and the Sahara. These are likely artifacts caused by the model's limited capabilities to capture the surface flask

measurements in those regions and are not reproduced by the satellite-only inversion.

## 1 Introduction

Carbon monoxide (CO) is toxic (Ryter et al., 2018) at high mixing ratios ($> 9$ ppm for an exposure of 8 h; much shorter at even higher mixing ratios, according to the World Health Organization (WHO, 1999)). However, CO mixing ratios in the





atmosphere are usually low enough that its toxicity and the resulting direct health effects are overshadowed by its indirect

effect on air quality. Most notably, CO is an ozone ($O_3$) precursor in the presence of nitrogen oxides ($NO_x$) and solar radiation (Holloway et al., 2000). The resulting tropospheric $O_3$ is again detrimental for humans and plants alike, even at low mixing ratios ($> 120$ ppb for an exposure of 1 h; or less for a longer exposure (Mckee, 1993)). Most CO will eventually be converted to carbon dioxide ($CO_2$) via reaction with the hydroxyl radical (OH) (Logan et al., 1981). As such, CO reduces the oxidative capacity of the atmosphere and both directly (by formation of $CO_2$) and indirectly (through the reduced OH abundance and

thus longer methane ($CH_4$) lifetime) increases greenhouse gas loads (Raub and McMullen, 1991; Daniel and Solomon, 1998; Heilman et al., 2014). As for the sources of atmospheric CO, almost half of it comes from the oxidation of methane and (Non-Methane) Volatile Organic Compounds (NM)VOCs, i.e. secondary CO production. The rest comes mostly from incomplete combustion of fossil fuels and biomass (e.g. wildfires or domestic wood burning), but also, in smaller quantities, from direct emissions from plants (biogenic CO) and the oceans (Zheng et al., 2019). While biomass burning makes up less than a quarter

of the total CO source in most years, those emissions come with the largest uncertainty (see Sect. 2.3.1 for more details), linked to their high spatial and temporal variability compared to the other sources.

Estimating regional CO emissions and dividing them up by source categories at a global scale is not trivial. While current remote sensing techniques allow for the observation of CO mixing ratios globally and at relatively high spatial and temporal resolutions, they carry insufficient information to directly infer the underlying emissions by source category. However, estimat-

ing CO emission sources indirectly is possible by incorporating some additional information in either bottom-up or top-down approaches.

In bottom-up estimates, the process that caused the emission is measured and from this the emissions are extrapolated. For example, if the cause of the CO emissions is a wildfire, emissions can be extrapolated based on knowledge about the burnt vegetation and the intensity of the fire. Conversely, in top-down estimates, the effect of the emissions is measured and traced

back to its source. Following again the example of wildfire CO emissions, the effect is an elevated CO concentration in the atmosphere, which can be observed and then traced back and attributed to its source using atmospheric modeling.

Both approaches are affected by various sources of errors. Bottom-up estimates usually require direct observations of the source event and some additional assumptions about the source itself, for example, fuel characterization (ecosystem type, fuel loadings, and fuel consumption rates) and emission factors in the case of biomass burning. Top-down estimates have more

loose observational requirements but require a set of potentially more elaborate assumptions for the atmospheric modeling, for example about chemistry and atmospheric transport. Overall, there is little overlap between the error sources and, therefore, one approach may be used to reduce the uncertainties of the other.

In this study, we use a top-down approach in the form of inverse modeling, specifically, the state-of-the-art inverse modeling framework TM5-4DVAR. Initial inversion studies using the global atmospheric chemistry transport model TM5 (Krol et al.,

2003) or the extended TM5-zoom (Krol et al., 2005) in combination with their respective adjoint versions can be found in Gros et al. (2003, 2004) for methyl chloroform and CO, and in Bergamaschi et al. (2005, 2007) for methane. The TM5-4DVAR inversion suit, as described in detail in Meirink et al. (2008b), is based on TM4-4DVAR (Meirink et al., 2006). A first application of TM5-4DVAR can be found in Meirink et al. (2008a). In this study, the CO branch (Krol et al., 2008) of



the TM5-4DVAR inversion suit is employed, which has been the basis for multiple other studies (Hooghiemstra et al.,
2011, 2012a, b; Krol et al., 2013; Nechita-Banda et al., 2018).

The basic concept of inversions in the TM5-4DVAR model is to modify a set of prior emissions (a priori) in a way that
minimizes the mismatch between the model and one or more sets of observations of atmospheric mixing ratios, to obtain an
optimized set of posterior emissions (a posteriori). By including information from additional observations, inverse modeling
is capable of reducing the uncertainties on the a priori emissions, which are usually taken from bottom-up inventories. The
observations used in inverse modeling can range from spatially and temporally sparse surface flask data (Bergamaschi et al.,
2000; Pétron et al., 2002; Butler et al., 2005; Pison et al., 2009; Hooghiemstra et al., 2011), over local aircraft measurements
(Palmer et al., 2003; Heald et al., 2004), to global satellite observations (Pétron et al., 2004; Arellano et al., 2004; Fortems-
Cheiney et al., 2009; Hooghiemstra et al., 2012a), or even combinations of multiple such datasets (Hooghiemstra et al., 2012b;
Krol et al., 2013; Jiang et al., 2017; Nechita-Banda et al., 2018).

Previous studies with the TM5-4DVAR model employed satellite observations from the Measurements of Pollution in the
Troposphere (MOPITT) instrument (Hooghiemstra et al., 2012a, b), the Infrared Atmospheric Sounding Interferometer (IASI)
instrument (Krol et al., 2013) or both (Nechita-Banda et al., 2018). In this study, we introduce a new satellite dataset into
the TM5-4DVAR inverse model, by using combined data from (a) the high-resolution TROPOspheric Monitoring Instrument
(TROPOMI) onboard the Sentinel-5 Precursor (S5P) satellite and (b) the NOAA surface CO flasks from the ESRL Global
Monitoring Laboratory and proposing an iterative process to more rigorously weight both datasets against each other in the
inversion. TROPOMI features several differences to, and advantages over MOPITT and IASI. Most notably, the TROPOMI
CO retrievals are performed solely in the short-wavelength infrared (SWIR, around $2.3\,\mu m$; Veefkind et al., 2012) range, as
opposed to IASI's mid-wavelength infrared (MWIR, around $4.76\,\mu m$; De Wachter et al., 2012) range. MOPITT uses mostly
the thermal MWIR bands around $4.6\,\mu m$, assisted by the solar SWIR band around $2.3\,\mu m$ (Drummond et al., 2010). By using
shorter wavelengths, the TROPOMI retrievals exhibit less interference from Earth radiation and are, therefore, more sensitive
to CO that resides close to the surface compared to MOPITT and IASI. Overall, TROPOMI has high sensitivity throughout the
atmosphere, whereas MOPITT and IASI are most sensitive to the middle and upper troposphere. However, the combination
with the SWIR band increases MOPITT's surface-level sensitivity under specific conditions (e.g. Worden et al., 2010). Fur-
thermore, TROPOMI procures CO observations at a high spatial resolution of up to $7 \times 7\,km^2$ (Veefkind et al., 2012), which
is roughly 10 times higher than the resolution of MOPITT (up to about $22 \times 22\,km^2$; Drummond et al., 2010) and the spatial
sampling of IASI (up to about $25 \times 25\,km^2$; Clerbaux et al., 2009). Additionally, TROPOMI takes one day to reach global
coverage, which is comparable to IASI, whereas the MOPITT instrument takes about five days to achieve the same.

However, the TROPOMI observations correspond to a large data volume due to their high resolution and high coverage,
which implies a large computational cost when using these data in the TM5-4DVAR inversion suit. One established way to
reduce the computational cost of global inversions is through zooming, where only a limited region is simulated at a fine reso-
lution, while the rest of the globe is simulated at a coarser resolution. This way, the everlasting trade-off, where increasing the
model resolution implies not only rising precision but also rising computational cost, can be partly overcome. This method has
been proven to yield very similar results within the limited fine resolution region compared to simulations with fine resolution





globally, while significantly reducing run times. Therefore, the coarser global simulation is still sufficient to provide mean-

ingful boundary conditions to the finer region of interest. Intermediate regions may be used to provide more fluent transitions between the coarse and the fine region. Such nested grids can be found for example in TM5-4DVAR (Berkvens et al., 1999; Krol et al., 2005), and GEOS-Chem (Wang et al., 2004; Chen et al., 2009).

Similarly, the resolution of satellite observations can be reduced by defining a grid and aggregating all observations within each cell of this grid into a single so-called super-observation (Eskes et al., 2003; Miyazaki et al., 2012; Boersma et al., 2016).

Here, we use a modified version of this super-observation approach to reduce the number of observations in the dataset, which in turn reduces the computational cost they introduce in the inversion.

In this study, we investigate the added value of the new TROPOMI data in constraining global CO emissions. As a proof of concept, we focus on the emissions in the northern hemisphere in the second half of 2018. We have split this investigation into a series of experiments, in which we run the same inversion multiple times, each time with slightly different settings. Firstly,

we optimize CO emissions simultaneously towards TROPOMI satellite observation gridded to $0.5° \times 0.5°$ and NOAA surface flask measurements. This inversion will be used as a reference case, against which all other inversions are compared. For this reference inversion, we will analyze the increments to the a priori emissions at the global scale, to identify short-comings in either the model or the bottom-up inventories that serve as a priori emissions. In the second step, we compare the reference inversion to two inversions where we vary the inventory used as biomass burning a priori emissions, to investigate the influence

of the a priori emissions. We focus on biomass burning emissions, since those have the largest uncertainty. Thirdly, we repeat the inversion with the same a priori emissions as in the reference case two more times, once with only the TROPOMI satellite observations (and no flask data) and once with only the NOAA flasks (and no satellite observations). Comparing the results of those inversions with the reference inversion gives insight into the impact of the TROPOMI observation on the inversion results by highlighting areas where satellite observations and station measurements carry unique, redundant or even conflicting

information. Finally, we also run the inversion with the full resolution satellite observations (up to $7 \times 7\,\mathrm{km}^2$) in combination with the NOAA surface flasks, to analyze the influence of gridded satellite observations on the model at its relatively coarse resolution of $3° \times 2°$.

## 2   Materials and methods

### 2.1   Model description

The Cycle 3 TM5-4DVAR model as of revision c71f31 from the official code repository of the model (https://sourceforge.net/p/tm5/cy3_4dvar/) is used. In the scope of this study, the existing code is extended to handle the high-resolution TROPOMI observations. Additionally, support for anthropogenic emissions based on CMIP6 is implemented, the capabilities to use the output from the full-chemistry model TM5-MP as initial conditions and as a priori for the secondary sources of CO are extended, and some minor compatibility issues are resolved. The specific code version used here is available at Nüß et al.

(2024a).



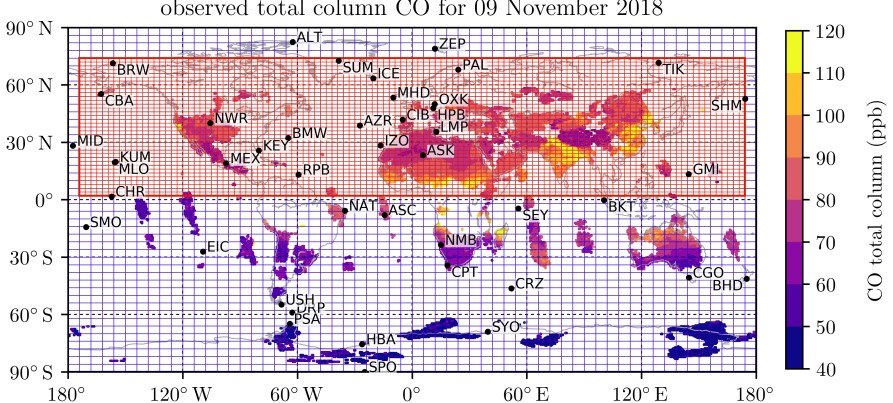

**Figure 1.** Used zooming setup, with $6° \times 4°$ grid globally (blue) and nested $3° \times 2°$ grid over the northern hemisphere (red). The locations of the background stations where the NOAA CO flask measurements are collected are shown as black dots and labeled with their respective station ID. The colormap shows the used global TROPOMI satellite observations for one day (9 November 2018) as an example of the daily coverage and resolution they provide. Due to strict quality filtering during the retrieval process (Schneising et al., 2023), many places have no valid TROPOMI observations, despite every location on Earth being visible to the instrument at least once per day. A more comprehensive overview of the TROPOMI CO data coverage for all of 2018 can be found in supplemental Figs. S1 and S2.

In the offline model TM5-4DVAR, atmospheric transport and chemistry are driven by preprocessed meteorological fields from the European Centre for Medium-Range Weather Forecasts (ECMWF) Re-Analysis project (ERA-Interim meteorology; Dee et al., 2011) coarsened to the lateral model resolution and 34 altitude layers (from surface pressure to the top of the atmosphere (fixed to 47.8 hPa in the top layer), with the highest resolution in the Upper Troposphere-Lower Stratosphere (UTLS)). Advection is calculated using the slope scheme developed by Russell and Lerner (1981). In that scheme, for each model box not only the tracer mass, but also three slope values are stored, to capture the gradients in north-south, east-west and up-down directions. These slopes increase when- and wherever tracer mass enters or leaves a cell and level out over time otherwise.

By employing the zooming technique described in Berkvens et al. (1999), the TM5-4DVAR model is capable of simulating only the region of interest at a high resolution (up to $1° \times 1°$; longitude $\times$ latitude), while the rest of the globe is simulated at a reduced resolution ($6° \times 4°$). In this study, the region of interest is simulated only at a medium resolution of $3° \times 2°$, but covers a very large area. The region of interest is placed over the northern hemisphere, spanning $2°\,\text{N}–74°\,\text{N}$ and $174°\,\text{W}–174°\,\text{E}$ and captures all major land masses, as shown in Fig. 1. The region of interest and the global region are two-way nested, i.e. at the beginning of each time step the finer region takes its boundary conditions from the coarser global region and at the end of each time step it also updates the coarser region with its more precise results.

In our inversions, we use the simplified CO-only chemistry version of TM5-4DVAR described in Hooghiemstra et al. (2011), which only explicitly considers the reaction of CO with OH. The OH is prescribed by the widely used monthly climatological fields from the TransCom-$CH_4$ project described in Patra et al. (2011), in which tropospheric OH is based on the OH fields





from Spivakovsky et al. (2000) scaled by 0.92, as suggested in Huijnen et al. (2010). In addition to the chemical loss due to

OH, CO also experiences loss due to dry deposition, which is simulated using the parameterization from Ganzeveld et al. (1998), adapted for TM5 and ERA-Interim meteorology.

## 2.2   4DVAR approach

The 4DVAR approach, which was first described and applied to meteorological assimilations by Talagrand and Courtier (1987), has been extended to assimilate atmospheric chemistry by Fisher and Lary (1995) and satellite data by Eskes et al. (1999). While

the first applications were strongly limited by computational power, the field flourished recently with rising computational capabilities and more extensive datasets. In the following, a quick rundown on the mathematical basis of the 4DVAR approach is provided, based on the more extensive description by Brasseur and Jacob (2017).

The aim of every inversion is to find the state $x$ (here the CO emissions) that fits best to the observations $y$ (here the CO columns from TROPOMI and surface flask observations from NOAA). To connect the state $x$ with the observations $y$,

the observational operator $\mathbf{F}$ is needed, which includes both the forward model and the spatial and temporal sampling of the observations:

$$y = \mathbf{F}(x, p) + \varepsilon_{\mathrm{O}}, \tag{1}$$

where $p$ are the model parameters, which is every input to $\mathbf{F}$ that is not part of the state $x$ (for example meteorology, a priori mixing ratios, or the used chemistry scheme) and $\varepsilon_{\mathrm{O}}$ is the observational error, i.e. the combined error of measurements, model,

and parameters.

Because $\varepsilon_{\mathrm{O}}$ is generally non-zero, there is no single trivial solution for $x$, that minimizes the difference between the right-hand and left-hand side of Eq. (1). Instead, the state $x$ has to be changed iteratively in a process called optimization. For each state $x$, a cost $J(x)$ can be defined, which provides information on how well that state fits the observations in a least-squares sense. Additionally, an a priori state $x_{\mathrm{A}}$ is required to regularize the otherwise ill-conditioned problem, preventing non-physical

behavior. This "initial guess" can be used to constrain the inversion to reasonable states, for example by not permitting biomass burning over the oceans. This leads to the cost function

$$J(x) = (x - x_{\mathrm{A}})^{\mathrm{T}} \mathbf{S}_{\mathrm{A}}^{-1} (x - x_{\mathrm{A}}) + (y - \mathbf{F}(x))^{\mathrm{T}} \mathbf{S}_{\mathrm{O}}^{-1} (y - \mathbf{F}(x)), \tag{2}$$

where $\mathbf{S}_{\mathrm{A}}$ and $\mathbf{S}_{\mathrm{O}}$ are the a priori and observational error covariance matrices, respectively.

During the optimization process, the model repeatedly runs forward and backward in time. During the forward run, the

mixing ratios at times and places of the observations are stored. Based on the stored model mixing ratios and the observations, the cost that corresponds to the current state $x$ can be calculated. During a backward run, the adjoint model, i.e. the adjoint of the tangent linear model, is used. In case of a linear problem, the tangent linear model is identical to the forward model. This adjoint integration is fed by the mismatches between forward model and observations (rather than tracer masses) and leads to the gradient of the cost function with respect to state vector elements $x$. Based on that gradient, the state (e.g. the

emission fields) for the next iteration cycle is adjusted, which then starts again with a forward run. This cycle is repeated until the gradient of the cost function is sufficiently reduced, i.e. the cost is close to its global minimum.





Overall, in 4DVAR, the model is sampled temporally and spatially for each individual data point, and each point provides its own contribution to the cost function. As such, this approach is well capable of assimilating multiple datasets with different spatial and temporal resolutions at once and co-sampling of observations across datasets is neither necessary nor detrimental.
One caveat is that the observations of different datasets need to be weighted properly against each other. On the one hand, this implies proper measurement error estimation. On the other hand, some form of error inflation (Sect. 3.2.2) might be required if datasets with vastly different numbers of observations are used, or if some datasets have a much higher resolution than the model.

In this study, the inversions are carried out using the non-linear M1QN3 optimizer described in Gilbert and Lemaréchal (1989). This optimizer is capable of handling a semi-exponential description of the probability density function for the a priori emissions, which in turn avoids negative emissions (Bergamaschi et al., 2009). As a convergence criterion, a reduction of the gradient norm of the cost function of $10^3$ is chosen, i.e. the iterations are stopped once the cost function is one thousand times less steep. This criterion was suggested in Meirink et al. (2008b) to be sufficient to converge the emissions. With this criterion, it takes the model around 35 iterations to converge, whereas the budget terms are near constant for the last few iterations.

## 2.3 Model setup

The TM5-4DVAR model, as described in Sect. 2.1, is used to perform multiple inversions of the CO emissions in the year 2018, with a specific focus on the northern hemisphere.

### 2.3.1 Inventories and emission categories

CO production from three distinct source categories – anthropogenic, biomass burning, and secondary CO production through chemistry – is considered. Since the contributions of oceanic and biogenic CO to the overall source are small compared to the aforementioned categories, they have been neglected in this study. Additionally, no daily cycles in emissions or chemistry were considered, mostly due to limitations of the OH climatology (see Sect. 2.1) and the secondary CO production a priori (introduced further down in this section).

As biomass burning a priori emissions we use the Fire INventory from NCAR version 2.5 (FINN2.5), which is described in Wiedinmyer et al. (2023) and available at Wiedinmyer and Emmons (2022). FINN is based on three data products from the Moderate Resolution Imaging Spectroradiometer (MODIS), namely those for active fires, land cover type, and vegetation continuous fields, which are used to infer burned area and fire emissions. Compared to the original FINN version 1 (Wiedinmyer et al., 2011), the FINN version 2 used in this study features an improved representation of large fires by merging overlapping fire pixel areas. Additionally, rather than using a single static vegetation map for all years, the respective MODIS land cover type and vegetation continuous field data from the previous year are used. Also, the fuel loadings and emission factors have been updated. Specifically, we use FINN2.5+VIIRS, which includes additional small fire detection via satellite observations from the Visible Infrared Imaging Radiometer Suite (VIIRS) and NMVOCs speciated to the Model for OZone And Related chemical Tracers (MOZART-T1) chemical mechanism (Emmons et al., 2020).



As a sensitivity study, we conduct additional inversions where we replace FINN2.5+VIIRS as the biomass burning a priori
with (1) FINN2.5 (without VIIRS), and (2) emissions from the Global Fire Emissions Database version 4, including small fire
boost (GFED4.1s; Randerson et al., 2017). The inversion experiments are introduced in more detail in Sect. 2.3.4.

GFED4.1s is based on satellite observations of burned area from MODIS, and fire activity from both the Visible and Infrared
Scanner (VIRS) and the Along Track Scanning Radiometer (ATSR; Giglio et al., 2013). These observations are combined with
datasets on vegetation characteristics and meteorology to infer burned area and fire emissions on monthly scales, along with
scaling factors to receive higher (daily or 3-hourly) temporal resolutions (van der Werf et al., 2017). The small fire boost
includes estimates for biomass burning emissions from fires that are below the detection limit of the burned area product
(MODIS), but are still visible as thermal anomalies (Randerson et al., 2012). While these estimates have fairly large errors on
a local scale (Zhang et al., 2018), including them leads to more realistic total biomass burning emissions on the regional to
global scale of the model used in this study.

Both GFED and FINN are coarsened to the resolutions of the zooming regions and aggregated into daily bins to serve as
global priors for the biomass burning emissions. After applying the emission factors, all fire types are lumped together into a
single biomass burning fire type. Since both inventories only provide 2D surface level emissions, they are used in conjunction
with injection heights from the IS4FIRES Integrated Monitoring and Modelling System for wildland fires developed at FMI
(Sofiev et al., 2012, 2013).

For calculating the contribution to the cost function, a grid-scale a priori error of 100 % is assumed globally for the biomass
burning emissions. This error is constructed from the error of at least 50 % provided in van der Werf et al. (2017) for the
regional carbon emissions in GFED4.1s, combined with the error of the emission factors that are used to convert the total
(carbon) emissions of each fire type into distinct species (e.g. $CO$). These are fixed per fire type and are reported with an
estimate of their natural variation in the order of one-third of the reported value (Akagi et al., 2011). Since GFED4.1s and
FINN2.5(+VIIRS) are fairly similar in terms of spatial distribution and amplitude of wildfire emissions (see Supplemental
Fig. S3, note the logarithmic scale) and to keep the inversion results comparable, we assume an a priori error of 100 % for
FINN2.5(+VIIRS) as well. Additionally, to prevent erroneous biomass burning emissions in the inversion result, the a prior
error is set to zero over the oceans. While this implies fixed biomass burning emissions for relatively small islands, for example
Hawaii, emissions from large islands, for example Indonesia, are still optimized.

TM5-4DVAR allows for spatial and temporal correlations for each emission category to be set. These reduce the effective
number of degrees of freedom of the inversion, which can help to prevent overfitting of the observations and lead to more
realistic results, while also reducing the number of iterations needed to reach convergence (Meirink et al., 2008b). The numeric
values for the spatial correlation lengths and temporal correlation times stated in the following are empirical and follow the
values provided in Krol et al. (2013) and Nechita-Banda et al. (2018), who used a similar setup with the same model. Biomass
burning events are usually fairly temporary, so a short exponentially decreasing correlation time of 0.1 months for emissions at
different times in the same grid cell is used. To account for the usually small spatial extent of biomass burning events (compared
to the coarse resolution of the model grid), we use an exponentially decreasing correlation length of only 200 km for emissions
at the same time in neighboring grid cells. The biomass burning emissions are optimized at a daily resolution in the state (i.e.





the optimizer can change the biomass burning emissions for each day separately, but it cannot change any potential diurnal
patterns) to best capture the high temporal frequency of the burning events and therefore maximize the distinction between the
biomass burning emissions and the other categories. Previous studies (e.g. Krol et al., 2013; Nechita-Banda et al., 2018) used
a 3-daily resolution in the state (i.e. the optimizer could change the emissions in 3-day chunks, but not the relative emission
distribution from day to day within each chunk) and in Krol et al. (2013) a sensitivity study with daily resolution was conducted
with mixed results.

Secondary CO production from the oxidation of $CH_4$ and other VOCs is based on 3D production fields from a simulation
of the full chemistry model TM5-MP with the extended MOGUNTIA chemical scheme described in Myriokefalitakis et al.
(2020) for the year 2018. This source is optimized with a fairly conservative a priori error of only 20 %. We expect fairly
gradual changes for this source in time. Therefore, we use an exponentially decreasing correlation time of 9.5 months for the
secondary CO production at different times from the same cell. Note that this rather restrictive correlation time does not limit
the model's ability to capture the seasonality of short lived VOCs like isoprene, since that seasonality is already included in
the prior production fields. Instead, it only limits how much the deviations from those prior fields may vary from month to
month. Similarly, spatial emission changes are also expected to be gradual for secondary production, due to the well-mixed
$CH_4$ background, leading to an exponentially decreasing correlation length of 1000 km. A monthly resolution in the state is
chosen for the secondary CO production, i.e. the optimizer can change it only once per month and the production is constant
over the course of that month. Choosing this much coarser state resolution compared to the daily resolution for biomass burning
emissions, makes it cheaper, with respect to the cost function, for the optimizer to capture the usually short time scale biomass
burning events with the intended emission category. With all of this combined, the low a priori error, low state resolution, and
large temporal and spatial correlation, we hope to reduce aliasing between the smooth fields of this category and the more
patchy biomass burning emissions. Conversely, since NMVOC oxidation can be quite local occasionally, this approach bears
the risk of capturing part of the secondary production in the biomass burning emissions, specifically when the NMVOCs are
emitted by fire activity.

Anthropogenic CO emissions are taken from the Climate Model Intercomparison Project 6 (CMIP6) inventory (Eyring et al.,
2016), specifically the SSP370 (Fujimori et al., 2017; Riahi et al., 2017; Gidden et al., 2019) projection dataset (Gidden et al.,
2018). Due to the low interannual variation of anthropogenic emissions compared to secondary CO production or biomass
burning emissions and the fairly up to date inventory (with historical data up to 2014 and projected data from 2015 onwards),
a conservative a priori error of 10 % is assumed, with the same monthly state resolution as for the secondary production.
Following the same argument as for secondary CO production, we use an exponentially decreasing correlation time of 9.5
months. Similarly, spatial changes in anthropogenic emission are expected to occur on the level of countries or economic
zones, leading to an exponentially decreasing correlation length of 2000 km. As for the biomass burning emissions, changes to
these anthropogenic emissions are restricted to land. Thus, shipping emissions are included in the inventory, but not optimized.





### 2.3.2 Simultaneous inversion of multiple emission categories

As mentioned in the previous section, anthropogenic emissions, biomass burning emissions, and the secondary CO production are optimized simultaneously, i.e. they are all part of the state vector $x$ (Sect. 2.2) and the optimizer could adjust any of them to minimize the cost function. This approach will inadvertently lead to some aliasing between the categories, despite the rigid choices for the a priori error, correlation length and time, and state resolution for the secondary production category. However, optimizing the biomass burning emissions on their own is not an option either, since this will force the model to represent any mismatches by adjusting the biomass burning emissions, even if these mismatches actually stemmed from flaws in the chemical production or anthropogenic a priori. This extreme form of aliasing leads to very poor convergence at the background stations, even when extremely high a priori errors are assumed. By using not only sparse flask data, but also the high coverage, high resolution TROPOMI observations, we might be able to better distinguish between the emission categories.

### 2.3.3 Initial conditions, spin-up, and main inversions

The initial tracer distribution is an important part of the inversion. Close to the starting date of the inversion period, the initial tracer distribution must fit the total columns and horizontal distribution of the observational datasets reasonably well. If there are considerable over- or under-estimations, the emission increments will be dominated by the efforts of the model to fix the offset in the mixing ratios. These additional emissions will mask the actual signal of the observations, i.e. by how much the a priori emissions differ from the true emissions. Additionally, the initial vertical CO distribution must be realistic, since the CO depletion and transport vary with altitude. Therefore, assuming too high initial mixing ratios in a layer with low transport and low loss will affect the model for a long time.

To accommodate this, the period of interest (the year 2018) is split into two separate inversions. The first period is a spin-up inversion to harmonize the global distribution of CO mixing ratios in the model with the observational datasets (see Sect. 3). Harmonizing the model and the observations, especially in remote regions where transport is slow, requires the model to be run over a prolonged period of time. Therefore, the spin-up inversion is run over multiple months, from 1 January 2018 to 1 July 2018. The second period is the main inversion, which uses the harmonized mixing ratios from the spin-up inversion as initial conditions. It is run from 1 June 2018 over seven months until 1 January 2019 and leads to the results of scientific interest presented in Sect. 4.

Note the one month of overlap in the inversion periods. This overlap is necessary, because emissions close to the end of the inversion period are verified by very few observations. Therefore, the final month of the spin-up inversion is considered as its spin-down period, during which confidence in the generated emissions and the resulting mixing ratios is diminished. Similarly, the final month of the main inversion, December 2018, should be considered as its spin-down period. The duration of this period was chosen based on the lifetime of CO of around two months (Raub and McMullen, 1991; Holloway et al., 2000). Hence, a snapshot of the mixing ratios from the final iteration of the spin-up inversion from 1 June 2018 is used as initial conditions for the main inversion.





**Table 1.** A priori emissions and observational setup for the conducted experiments.

| Inversion | | | A priori emissions | | | Observations | |
|---|---|---|---|---|---|---|---|
| | | | biomass burning | anthrop. | secondary | satellite | flasks |
| *spin-up* | | | FINN2.5+VIIRS | | | gridded | yes |
| Main inversions | Set 1 | *reference* | FINN2.5+VIIRS | CMIP6 | TM5-MP | gridded | yes |
| | | *noVIIRS* | FINN2.5 | | | gridded | yes |
| | | *GFED* | GFED4.1s | | | gridded | yes |
| | Set 2 | *satellite only* | FINN2.5+VIIRS | | | gridded | no |
| | | *stations only* | FINN2.5+VIIRS | | | none | yes |
| | | *full satellite* | FINN2.5+VIIRS | | | full | yes |

The spin-up inversion itself is started with tracer fields taken from the chemistry transport model TM5-MP, which employed the MOGUNTIA chemistry scheme. In Myriokefalitakis et al. (2020) and sources therein, a detailed description of the model, setup, and chemistry scheme, alongside extensive validation versus observational data can be found. In addition to the simulation analyzed and described there, the TM5-MP model was run with the same settings for a longer time frame, including 2018. Here, we use instantaneous concentrations from that simulation as initial conditions for the spin-up, and monthly chemical budget terms for the secondary source of CO from VOC oxidation.

The validations in Myriokefalitakis et al. (2020) have shown that the TM5-MP model generally produces reasonably realistic tracer fields in terms of both vertical and horizontal distributions. However, some offsets to the observations still remain. For CO specifically, Myriokefalitakis et al. (2020) found too low mixing ratios in the northern hemisphere and too high mixing ratios in the southern hemisphere. The spin-up inversion in this study is necessary to confidently remove those offsets.

Additionally, the spin-up inversion facilitates a smooth transition between the different emission datasets used by Myriokefalitakis et al. (2020) in TM5-MP and in this study in TM5-4DVAR. While we both use CMIP6 for anthropogenic CO and the same meteorology, they use CMIP6 also for biomass burning, while we use FINN2.5 or GFED4.1s. We do that because, for 2018, both these inventories provide historical data rather than projections and inversions strongly benefit from realistic lateral a priori distributions, which cannot be obtained from projection data as those in CMIP6. Another important difference is the handling of OH. While their OH is calculated online, we use prescribed OH as described in Sect. 2.1.

### 2.3.4 Experiments

Table 1 gives an overview of the experimental setups for the inversions analyzed in this study. The main inversion period (1 June 2018 to 1 January 2019) is chosen based on the availability of the used input data and computational constraints. Regarding the input data, TROPOMI was in its commissioning until March 2018 and the ERA-Interim meteorology dataset ends in August 2019. The latter constraint will be lifted for future studies by switching to ERA5 meteorology (Hersbach et al.,





2020). Still, the large zooming region over most of the northern hemisphere, which is chosen to gain deeper insight into the general anthropogenic emission patterns, combined with the long inversion period come at a high computational cost. Each inversion takes about five real-world days to run (even longer with the full resolution satellite observations). Therefore, the inversion period does not extend into 2019. Emissions for this period are optimized a total of six times with different settings, split into two sets.

In the first set, we vary the biomass burning a priori emissions, while using the same observations (global gridded TROPOMI observations in conjunction with flask measurements from the NOAA background stations) to constrain the emissions. More details on the a priori emission inventories and the observations used, including the gridding process, can be found in Sects. 2.3.1 and 3, respectively. With these inversions we intend to investigate the sensitivity of the optimized emissions to the a priori, since we introduce a new and updated version of FINN into the model and apply a significantly lower grid-scale biomass burning a priori error compared to previous studies. The first set includes (1) the *reference* inversion with FINN2.5+VIIRS, (2) the *noVIIRS* inversion with regular FINN2.5 and (3) the *GFED* inversion with GFED4.1s.

In the second set, the biomass burning emissions are kept fixed to the *reference* case (FINN2.5+VIIRS) and the observational datasets are varied. This way, we can assess the information content in the different datasets and the loss of information through gridding. The second set includes (4) the *full satellite* inversion using the full resolution satellite data in conjunction with the NOAA surface flasks, (5) the *satellite only* inversion using only the gridded satellite observations but no surface flasks and (6) the *station only* inversion using no satellite observations at all, where the inversion is driven solely by the surface flasks.

For the *spin-up* inversion (1 January 2018 to 1 July 2018) we use the same setup as for the *reference* inversion, i.e. FINN2.5+VIIRS as biomass burning a priori and gridded satellite observations in conjunction with NOAA surface flasks. All of the main inversions are started from this one *spin-up*, to ensure comparability of the results.

## 3 Observations

### 3.1 In situ measurements

The in situ observations used here are the NOAA surface flask CO measurements from various stations assembled by the Carbon Cycle Greenhouse Gases (CCGG) group (Petron et al., 2020). For filtering out non-background stations, the algorithm described in Hooghiemstra et al. (2012a) is applied to the 54 stations active between January and December 2018. Following this, only the 44 stations shown in Fig. 1 are classified as background and subsequently used. This filtering is necessary to avoid the large representation error introduced by non-background stations. On the one hand, the model has a fairly low resolution and will not be able to capture local sources that might affect the stations. On the other hand, it also has a relatively short time-step compared to the weekly or even bi-weekly station measurements, which is why a daily cycle may be caught by the model but not by the stations. Therefore, any station where the model shows a large diurnal cycle is excluded. The criterion is a mean daily standard deviation of more than $3.5\,\mathrm{ppb}$, following the example of Hooghiemstra et al. (2012a). However, background stations and those affected by seasonal biomass burning signals are kept; in other words, large annual standard deviations are allowed. Using only background stations comes with the implied assumption that air masses reaching them are well-mixed





and, therefore, even the coarse resolution of the model ($6° \times 4°$) is sufficient to capture the remaining spatial and temporal variation, allowing for a proper direct comparison of the model to the point observations. To account for any discrepancies from this assumption, the model estimates a representation error for each station based on the slopes (slope scheme introduced

in Sect. 2.1) in the box that contains the station.

For the station data, in addition to the representation error of the model, a sampling error of 2 ppb is assumed. This error is composed of the instrument precision of 1.5 ppb given in Gerbig et al. (1999) for the fast-response vacuum-UV resonance fluorescence CO (VURF) instrument used at all stations in 2018 and the reproducibility of the measurements of 0.5 ppb provided in the readme of the dataset (Petron et al., 2020).

**3.2   Satellite observations**

The second assimilated dataset consists of the CO total columns from the TROPOspheric Monitoring Instrument (TROPOMI) on-board Sentinel-5 Precursor (S5P) satellite launched in October 2017 (Veefkind et al., 2012). TROPOMI provides daily global coverage with a local overpass time at 13:30. The retrieved CO columns also feature a high spatial resolution of up to $7 \times 7\,\mathrm{km}^2$ at a swath width of 2600 km. Compared to that resolution, even the finest resolution of the model of $1° \times 1°$ might

seem very coarse. However, using high resolution observations not only implies a reduced aggregated observational error if multiple observations are available in a single model grid box, but it also gives a chance of at least some cloud-free pixels, i.e. some information, in cloudy model grid boxes.

For this study, we use the TROPOMI/WFMD version 1.8 product from the Carbon and Greenhouse Gas Group at the Institute of Environmental Physics (IUP) of the University of Bremen, retrieved with the Weighting Function Modified Differential

Optical Absorption Spectroscopy (WFM-DOAS) algorithm, which is described and validated in Schneising et al. (2019, 2023). This retrieval makes use of the TROPOMI observations in the shortwave infrared (SWIR) 2.3 μm spectral range to provide column-averaged dry-air mole fractions of methane and CO. The resulting total columns feature nearly constant sensitivity with respect to altitude. Notably, this includes the troposphere and boundary layer, which is especially useful when investigating biomass burning events and tropospheric air quality. In addition, observations in the SWIR spectral range, unlike those based

on visible light, are capable of seeing through smoke plumes to some degree, making them critically valuable for investigating biomass burning events. The latter works for smoke but not clouds due to vastly different particle sizes, as demonstrated in Schneising et al. (2020).

As detailed in Schneising et al. (2023), the retrieval employs a fairly strict quality filter, especially with regard to cloudiness, surface brightness, and solar zenith angle ($< 75°$). This selection implies a clear sky bias in the observations, resulting in an

overestimation of photochemical conditions, as well as very sparse data over the oceans due to their low albedo. The latter can be seen in Fig. 1, where over the oceans observations are only possible due to sun glint, which occurs almost exclusively in the center of the orbits (i.e. in a nadir viewing geometry), while the sun is at the zenith. This implies that the sparse observations over the oceans are mostly clustered together.





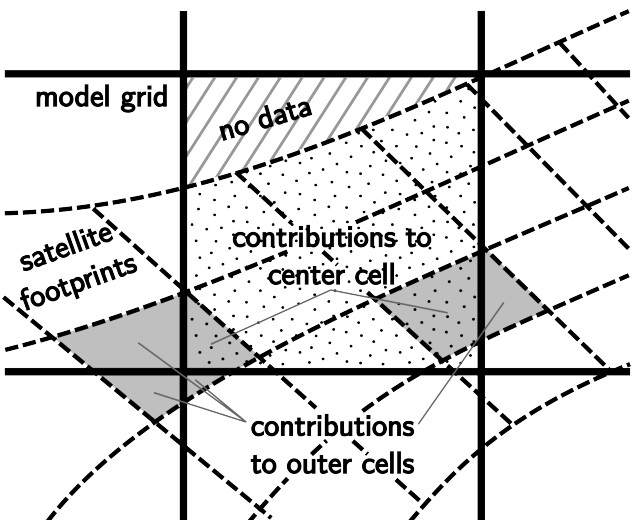

**Figure 2.** Schematic representation of several satellite footprints (outlined with dashed lines) intersecting with cells of a regular grid (thick, solid lines). The dotted areas show the portion $w_i$ of each footprint that contributes to the center grid cell with area $A_\text{cell}$. For footprints that intersect with more than one grid cell (two examples highlighted in grey), their contributions are further deweighted based on the ratio between their respective intersecting area $w_i$ (i.e. the part that is both dotted and grey) and their total area $A_i$ (the entire grey area). For the striped area no observations are available, hence, the coverage $\alpha$ for the center cell is $< 1$.

### 3.2.1 Gridding

Above, inversions with gridded satellite observations were referenced. To create these so-called super-observations, we follow the approach outlined in Miyazaki et al. (2012). As shown in Fig. 2, for each orbit, we calculate the intersection areas $w_i$ of the footprint of each observation $\hat{y}_i^\text{o}$ with the cells of a regular $0.5° \times 0.5°$ grid. We chose this grid resolution based on sensitivity studies conducted in our group (unpublished data), which have shown that at the coarse model resolutions used in this study, inversions based on observations gridded to $0.5° \times 0.5°$ lead to almost the same optimized emissions as those based on the

full satellite data, but with a significantly reduced computational cost (using full satellite data entails roughly 25 % longer computation times per iteration). According to Miyazaki et al. (2012), a representative super-observation for each orbit and grid cell can be calculated as an area-weighted average:

$$\hat{y}_\text{o} = \frac{\sum_{i=1}^m w_i \hat{y}_i^\text{o}}{\sum_{i=1}^m w_i}, \tag{3}$$

where $m$ observations contribute to this super-observation.

Notably, this average is not weighted by the retrieval error, which stems from the nature of the retrieval, where larger values have larger (absolute) errors, and, therefore, an error-weighted average would be skewed towards low values, as explained in Boersma et al. (2016). The same process of calculating area-weighted averages is also applied to the measurement time, the a priori profile, the pressure levels of the retrieval, and the averaging kernel, level-wise for the latter three.





Unlike Miyazaki et al. (2012), before calculating the super-observation error as an area-weighted average, we first inflate the error corresponding to each individual intersection $w_i$ so that its weight in the cost function (Eq. (2)) does not depend on the number of grid cells the corresponding footprint intersects with. This independence can be achieved with a factor $\sqrt{\frac{A_i}{w_i}}$, where $A_i$ is the total area of the satellite pixel's footprint, which contains the $i$-th intersection. The area $A_i$ is equal to $w_i$ if the footprint intersects exactly one grid box. Otherwise it will be larger, as exemplified in Fig. 2, where the areas $A_i$, highlighted in grey, are larger than the areas $w_i$ that are simultaneously grey and dotted for the two example footprints. The root stems from the least-squares nature of the cost function, while the rest is simply the inverse of the fraction of the footprint that intersects with the current grid cell. Taken together this yields an area-weighted error:

$$\sigma = \frac{\sum_{i=1}^{m} \sqrt{\frac{A_i}{w_i}} w_i \sigma_i^{\mathrm{o}}}{\sum_{i=1}^{m} w_i} = \frac{\sum_{i=1}^{m} \sqrt{A_i w_i} \sigma_i^{\mathrm{o}}}{\sum_{i=1}^{m} w_i}. \tag{4}$$

Further following Miyazaki et al. (2012), this $\sigma$ is then deflated by the number $n$ of observations that contribute to the super-observation in that grid cell. However, this deflation is limited by the correlation $c$ between errors of the individual observations (i.e. systematical errors from e.g. the albedo assumed in the retrieval are correlated in space and do not average out) as suggested in Eskes et al. (2003), and therefore, the super-observation error can be estimated as:

$$\sigma_{\mathrm{o}} = \sigma \sqrt{\frac{1-c}{n} + c}. \tag{5}$$

Exact values for $c$ are difficult to obtain, however, an upper bound may be found by considering the ratio of the systematic error of the TROPOMI observations versus its random error. From the validations against other observational datasets in Schneising et al. (2023), this ratio can be estimated to be roughly 30 %. As not all systematic error sources from observations within each $0.5° \times 0.5°$ grid box are correlated, $c = 15\,\%$ is assumed here. It should be noted that the exact value of $c$ has nearly no influence on the final inversion results, because a larger (smaller) $c$ leads to overall larger (smaller) errors, which, for the most part, will be canceled out by a larger (smaller) error inflation (Sect. 3.2.2).

However, this $\sigma_{\mathrm{o}}$ does not yet include the representativeness error, which accounts for potential differences between the true average tracer concentration (which includes the parts of the cell that are not covered by observations) and the $\hat{y}_{\mathrm{o}}$ calculated above. For example, if the satellite observes a pristine background in one part of the grid cell, but there is also a plume with high tracer concentrations obscured by clouds in the remaining area, $\hat{y}_{\mathrm{o}}$ would be too low. The more of the grid cell area is covered, the smaller this representativeness error becomes.

Miyazaki et al. (2012) suggest a method to estimate this effect. First, the initial mean observation in a cell and the coverage $\alpha = \frac{\sum_{i=1}^{m} w_i}{A_{\mathrm{cell}}}, 0 \leq \alpha \leq 1$, where $A_{\mathrm{cell}}$ is the total area of the grid cell, are calculated. In Fig. 2 the $\sum_{i=1}^{m} w_i$ is the total dotted area, whereas the $A_{\mathrm{cell}}$ is the total cell area enclosed by the thick, solid lines. Next, for well covered grid cells ($\alpha > 90\,\%$ in Miyazaki et al. (2012)), the coverage $\alpha$ is artificially reduced by randomly removing observations. For each observation removed, the mean and coverage of the remaining observations are recalculated. The new mean is then compared to the original value to yield a relative deviation. By repeating this process for many grid cells, a mean relative deviation $f_{\mathrm{rep}}(\alpha)$ can be calculated. Multiplying this relative deviation with the super-observation value $\hat{y}_{\mathrm{o}}$ gives the representativeness error





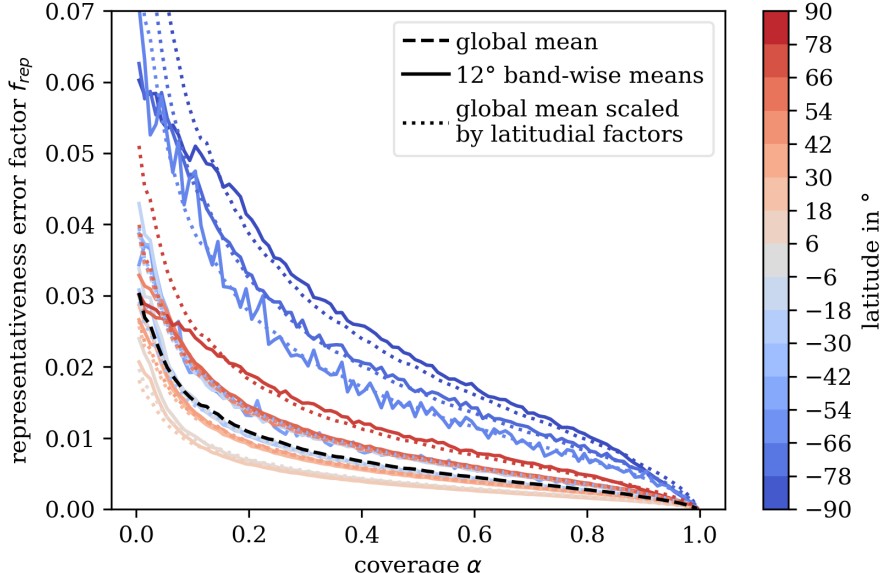

**Figure 3.** The dashed black line shows the global mean representativeness error factors over the satellite coverage in a given grid cell. This factor is zero for full coverage ($\alpha = 1$) and sharply increases at low coverage values. The colored lines show the mean representativeness error factors over 12° bands. As these are quite noisy, we instead use them to obtain a single scaling factor for each band. These factors are then multiplied onto the global mean representativeness error factors, which leads to the much smoother colored dotted lines.

for that cell. In Miyazaki et al. (2012), the mean observations are calculated as a simple arithmetic mean, whereas we use the area-weighted average introduced above:

$$f_{\text{rep}}(\alpha_k) = \left| \frac{\hat{y}_\text{o} - \frac{\sum_{l=1}^{m-k} w_l \hat{y}_l^\text{o}}{\sum_{l=1}^{m-k} w_l}}{\hat{y}_\text{o}} \right|, 0 < k < m, \tag{6}$$

where $k$ are the removed observations. For the sake of this analysis, we treat the initial observations in each grid cell, i.e. before removing any of them, as if they fully covered the cell. Therefore, $\alpha_k = \frac{\sum_{l=1}^{m-k} w_l}{\sum_{i=1}^{m} w_i}$ is the coverage compared to the initially covered area, rather than the full grid cell area.

In this study, to estimate the representativeness error, we analyze 31 days of data, evenly spread over the available observations for 2018. Additionally, we relax the coverage requirement to 50 % to have a larger set of eligible observations, especially when considering coarser grids (not shown in this study). As $\alpha_k$ is a continuous variable, we decided to aggregate it into 1 % bins for the sake of calculating the mean $f_{\text{rep}}(\alpha)$ over the entire analyzed data. The resulting global mean representativeness error is shown as the black dashed line in Fig. 3.

We noticed a weak intra-annual variation in the representativeness error factor, with generally slightly larger error values in the northern hemispheric summer. However, its magnitude was smaller than the temporal variation on a daily basis. Therefore, we decided to keep the representativeness error fixed in time.



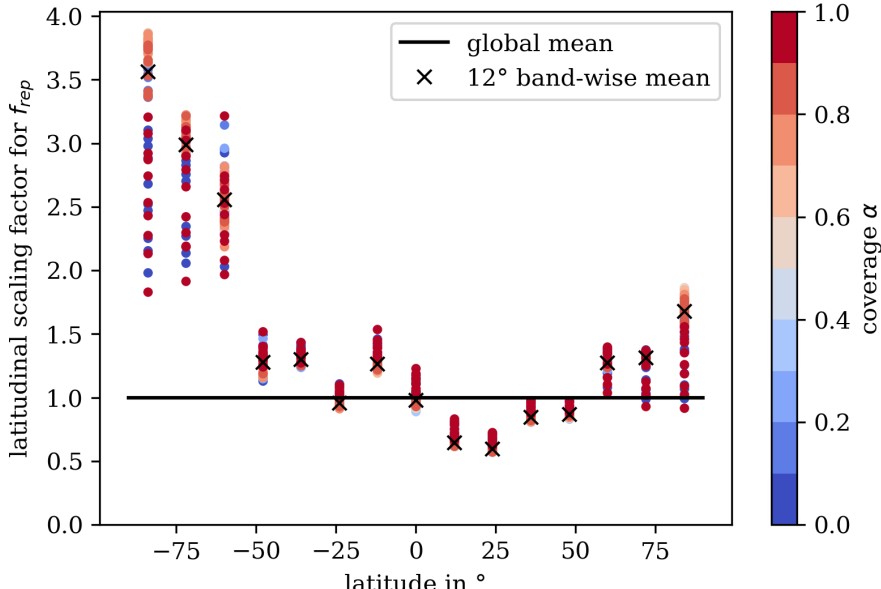

**Figure 4.** The black crosses are the $12°$ band-wise scaling factors for the global mean (black line) representativeness error factors, as shown in Fig. 3. Clearly, representativeness errors rise towards the poles, especially in the southern hemisphere where there is less land-cover. Additionally, the band-wise scaling factors for each $1\%$ coverage bin, normalized over the respective global mean for that bin, are shown as colored dots.

In latitudinal direction, we disregard the very few observations with a center point beyond $89.93°$ north/south, as these might touch and reach beyond the poles, which is problematic for area calculations in the latitude-longitude projection we employ. Additionally, as can be seen exemplified by the colored lines in Fig. 3, there seems to be a strong latitudinal dependence of the representativeness error, with larger values towards the poles and in the southern hemisphere. This latitude dependence is likely caused by the poorer measurement quality over the oceans and in high latitudes, and smaller grid cell sizes towards the

poles. Notably, while the magnitude of the representativeness error increases, the general dependence on the coverage $\alpha$ does not change. To capture this behavior, we additionally average the representativeness error factor over $\alpha$ for each latitudinal $12°$ band to obtain another scaling factor $\bar{f}_{\mathrm{rep}}(\phi)$, with $\phi$ as latitude. In Fig. 4, these band-wise factors are plotted before (colored dots) and after (black crosses) averaging over $\alpha$, all normalized over the global mean. With this, our total representativeness error factor is:

$$f_{\mathrm{rep}}(\alpha,\phi) = \bar{f}_{\mathrm{rep}}(\phi) \cdot f_{\mathrm{rep}}(\alpha) \tag{7}$$

The resulting latitude-wise representativeness error factors are shown as colored dotted lines in Fig. 3. The representativeness error can now be obtained for a given mean observation $\hat{y}_{\mathrm{o}}$, coverage $\alpha$ and latitude $\phi$ as

$$\sigma_{\mathrm{r}} = f_{\mathrm{rep}}(\alpha,\phi) \cdot \hat{y}_{\mathrm{o}}. \tag{8}$$



This leads to the total error of the super-observations

$$\sigma_{\mathrm{s}} = \sqrt{\sigma_{\mathrm{o}}^2 + \sigma_{\mathrm{r}}^2}. \tag{9}$$


The super-observations are always assumed to be located at the center of their corresponding cells. This might lead to a spatial bias, because observation within an arbitrary grid cell cannot generally be assumed to be evenly distributed.

### 3.2.2 Error inflation

The uncertainties provided for the individual satellite observations (for the *full satellite* inversion) and the total error of each of
the super-observations (for the inversions that use gridded satellite observations) are inflated with a global factor that depends on the specific inversion setup. For each inversion, this inflation factor is chosen so that the satellite and station observations each make up roughly half of the total observational cost, as suggested in Hooghiemstra et al. (2012a). The intent of this inflation factor is to capture the spatial correlation between the individual satellite footprints and to prevent them from suppressing the signal of the surface stations by their sheer number.

In previous studies, this inflation factor has only been roughly estimated. For example, an empirically chosen variance inflation of 2 was used in Chevallier (2007) for Orbiting Carbon Observatory (OCO) $CO_2$ observations gridded to $3.75° \times 2.5°$, an inflation of 50 was used in Hooghiemstra et al. (2012a) for MOPITT V4 level 3 CO observations gridded to $1° \times 1°$, and an inflation of again 50 was used in both Krol et al. (2013) and Nechita-Banda et al. (2018) for IASI CO observations at their native sampling resolution of up to about $25 \times 25\,\mathrm{km}^2$. Here, we suggest a more rigorous approach to finding the inflation that
fulfills the condition of having each dataset make up an equal part of the observational cost.

Finding the inflation factor at which this condition is fulfilled is in itself an iterative process, where each iteration is a complete inversion. A close look at the cost function (Eq. (2)) reveals that for an attempted inflation $I$, the inflation $I'$ for the next iteration can be calculated as

$$I' = \sqrt{\frac{J_{\mathrm{obs,sat}}}{J_{\mathrm{obs}} - J_{\mathrm{obs,sat}}} \cdot I^2}, \tag{10}$$

where $J_{\mathrm{obs}}$ is the total observational cost of the attempt, $J_{\mathrm{obs,sat}}$ is the part of $J_{\mathrm{obs}}$ contributed by the satellite observations, and the inflation factors $I, I'$ are a factor applied to the observational errors (standard deviations). It should be noted, however, that Eq. (10) will always underestimate the change in inflation needed. For example, if the initial inflation was too large, the formula will suggest an improved, but still slightly too large inflation for the next iteration. This happens, because reducing the inflation will increase the cost attributed to the satellite observations, which in turn causes the inversion to improve their
fit. However, a closer fit to the satellite observations usually implies degradation of the fit to the flask observations, which will increase their contribution to the cost function. That way, the total cost increases and a slightly smaller inflation is needed so that the contribution of the satellite observations makes up half of that cost. In the opposite case, if the inflation was too small, the next guess will be better but still slightly too small.

It may seem that the inflation is solely a parameter of the observational datasets involved and, therefore, fixed for a given set
of observations. However, we observed that the inflation also depends on the time of year, the error and temporal resolution





of the a priori emissions, and the a priori datasets used. Both, a larger a priori error or a higher temporal resolution of the emissions, especially for the biomass burning emissions, enable the model to fit the satellite observations more easily (lower cost) without degrading the station fit, leading to lower required inflation factors to fulfill the criterion.

With the setup outlined above, we obtained different inflation factors for the individual inversions. Inflation factors are generally larger for the main inversions compared to the *spin-up* inversion (42). Among the main inversions, we found slight differences based on which biomass burning prior was used. The inflation factors are largest for the *reference* inversion (64), followed by the *noVIIRS* inversion (63), and smallest for the *GFED* inversion (62), possibly due to smaller a priori mismatches at the stations, as elaborated later. Due to using the same emission setup, the *stations only* and *satellite only* inversions use the same inflation factor as the *reference* inversion, to maintain a similar weight of their background costs to their observational costs and for any analysis steps that require this value to be defined. These (standard deviation) inflation values are larger than the aforementioned variance inflation factors used in Hooghiemstra et al. (2012a) for gridded MOPITT observation, and in Krol et al. (2013) and Nechita-Banda et al. (2018) for full resolution IASI observations. The larger values are expected, because of the higher grid resolution when compared to MOPITT, and the better coverage of TROPOMI when compared to IASI. Due to the much larger number of observations, the largest inflation is required for the *full satellite* inversion (164). This number is an indication of the higher spatial correlation within the individual observations compared to within the gridded observations, since the latter are, by definition, further apart.

The concentrations at the locations of the surface stations depend only relatively weakly on the exact value of the inflation factor, because the well-mixed background concentrations show much broader patterns, which are captured by either dataset to some extent. However, very small inflation factors will still cause the station fits to degrade heavily, because the satellite data will drown out the flasks. Conversely, for very large inflation factors the model approaches the *station only* inversion. This emphasizes the need for the inflation factor to properly weigh both datasets against one another.

However, we concede that there are some issues with the condition of having the observational cost equally distributed between the stations and the satellite observations. This condition implies that satellite observations with higher coverage or lower errors are assigned higher inflation values, i.e. higher quality data gets a lower weight in the cost function. Inadvertently, this will lead to overfitting of the surface flasks with increasing quality of the satellite instruments used. Additionally, while we do expect a somewhat larger inflation at higher coverage due to increased correlation between the individual pixels, the current blanket approach of assigning a constant inflation factor to all footprints ignores the actual density and correlation of the observations. This implies that dense observations over the Sahara are inflated just as much as the sparse observations over the oceans. For future studies, this weighting strategy may need to be revised.





**Figure 5.** Modeled a priori (dotted lines) and a posteriori (solid lines) mixing ratios sampled at the locations of the stations as well as the flask observations (blue crosses) for six example stations and the three different biomass burning a priori inventories. For each observation, the corresponding measurement error is indicated as well. Lines are color-coded based on the a priori used: FINN2.5+VIIRS (*reference*) in orange, FINN2.5 (*noVIIRS*) in green and GFED4.1s (*GFED*) in pink. Unlike the first four, the bottom two stations ((e) PSA and (f) EIC) are in the southern hemisphere and, therefore, in the low resolution global region.





# 4 Results

## 4.1 Mixing ratio mismatch at the surface stations

### 4.1.1 Set 1: Inversions using different biomass burning priors

In Fig. 5, the modeled mixing ratios at 6 out of the 44 total ground-level stations are shown before and after the inversions from the first set of experiments (*reference*, *noVIIRS*, and *GFED*), where the biomass burning inventories were varied. Additionally, the corresponding flask measurement values as well as their assigned uncertainties are indicated. During the *spin-up* inversion (not pictured), many stations initially exhibit considerable under- or overestimations. The model corrects most of these within the first one or two months and the mixing ratios at the stations start to closely follow the observations. This way, during the main inversions (e.g. as shown in Fig. 5), the modeled mixing ratios at all stations are initially close to the observations. At most stations, the mixing ratios simulated based on the optimized emissions remain close to the observations over the whole period of the main inversion. This can be seen for example at Mauna Loa (Fig. 5d) and Rapa Nui (Fig. 5f) in the northern and southern Pacific, respectively, but also at stations close to the South Pole, like Palmer Station in Fig. 5e, despite their very remote nature.

However, at a few stations, the posterior mixing ratios diverge from the measurements to some degree. This effect is mostly limited to high ($> 55°$ N) northern latitudes. For example at Alert, as shown in Fig. 5a, mixing ratios in July and August do not drop far enough, while towards the end of the year they do not rise high enough. Another problematic station is Assekrem, plotted in Fig. 5b, where the flask observations are systematically underestimated by the model.

Generally, the a priori mixing ratios feature a global accumulation of ground-level CO over time not supported by the observations. This indicates an unbalanced budget, with either too large sources (overestimations in the a priori), or a too small sink (underestimations in the OH climatology). Given the setup of the inversions, the model resolves this by reducing the emissions in either case. However, there are stations where this does not hold and the a priori underestimates the observations. For example at Hohenpeissenberg in Fig. 5c, the model finds a fairly strong diurnal cycle and generally too low a priori mixing ratios. The former is likely a result of the station being located at the top of a mountain, where upslope conditions cause surface CO to be transported up to the station during daytime and away during night. Even though not clearly visible in Fig. 5c, where the full time series is shown, the model is only sampled at the time of the measurement, which would alleviate this issue to some degree. The too low a priori mixing ratios, however, could point to the relative proximity of the station to emission sources in Central Europe, and possibly indicate that the lateral model resolution is not fine enough to properly capture this station.

In the first eight rows of Table 2, we calculated the mean error-weighted mismatch $\bar{J}_{\text{flask}}$ between flasks and model for all main inversions, as

$$\bar{J}_{\text{flask}}(\boldsymbol{x}) = \frac{\sum_{i=1}^{N_{\text{flask}}} \left[ \frac{(y_{\text{flask},i} - \mathbf{F}(\boldsymbol{x})_i)^2}{\varepsilon_{\text{O},i}^2} \right]}{N_{\text{flask}}}, \tag{11}$$

where $N_{\text{flask}}$ is the total number of flask measurements $y_{\text{flask}}$ with observational errors $\varepsilon_{\text{O},i}$, and $\mathbf{F}(\boldsymbol{x})_i$ is the model sampled at that measurement. The observational errors include the representation error of the model and the sampling error of the flasks.





**Table 2.** Error-weighted mismatches between observations and model for all main inversions. The first eight rows give the mean mismatches to different subsets of the flask measurements. There, even in the *satellite only* inversion, where the flasks did not constrain the emissions, the overall fit at the stations improves, although less compared to the other experiments. The mismatch for the *satellite only* inversion decreases significantly if only stations south of $55°$ N are considered (i.e. excluding ALT, BRW, CBA, ICE, PAL, SUM, TIK, and ZEP), while it stays roughly the same for all other experiments. A considerable portion of the remaining mismatch stems from the stations ASK, HPB, and OXK, where the model generally has problems capturing the modeled variation. The last two rows contain the total mismatch to the satellite observations, scaled down by $10^3$ for readability. Similarly to the *satellite only* inversion above, even in the *station only* inversion, the overall fit to TROPOMI improves, despite those observations not constraining the inversion.

| | observations | | reference | noVIIRS | GFED | satellite only | station only | full satellite |
|---|---|---|---|---|---|---|---|---|
| stations | all | prior | 20.58 | 18.18 | 15.87 | 20.58 | 20.58 | 20.58 |
| | | poste | 3.69 | 3.92 | 3.99 | 9.29 | 3.31 | 3.57 |
| | $< 55° N$ | prior | 22.93 | 20.11 | 16.52 | 22.93 | 22.93 | 22.93 |
| | | poste | 3.66 | 3.86 | 3.97 | 7.87 | 3.41 | 3.57 |
| | exlc. ASK, HPB, OXK | prior | 20.75 | 17.90 | 15.63 | 20.75 | 20.75 | 20.75 |
| | | poste | 3.45 | 3.67 | 3.68 | 7.87 | 3.19 | 3.35 |
| | $< 55° N$ and exlc. ASK, HPB, OXK | prior | 23.35 | 19.92 | 16.26 | 23.35 | 23.35 | 23.35 |
| | | poste | 3.35 | 3.54 | 3.56 | 5.93 | 3.25 | 3.29 |
| | satellite | prior | 89.85 | 75.34 | 64.50 | 89.85 | 89.85 | 71.85 |
| | | poste | 8.14 | 8.51 | 8.66 | 7.07 | 20.80 | 7.79 |

If the model is capable of capturing the variability of the observations, the unit-less quantity $\bar{J}_{\text{flask}}$ should be close to one. Larger values could point to an underestimated observational error, systematic errors in the model itself, or a model with too few degrees of freedom to capture the variability in the observations, i.e. an underestimated model representation error. When

comparing two inversions, lower values represent a better fit. As can be seen for all three experiments of the first set (*reference*, *noVIIRS*, and *GFED*), the fit after the inversion is vastly improved compared to the prior fit. Considering how well the model captures the variability at most stations (e.g. Fig. 5), the a posteriori $\bar{J}_{\text{flask}}$ values of 3 to 4 most likely indicate underestimated errors, rather than systematic model errors. Table S1 in the supplement provides the individual mean error-weighted a priori and a posteriori mismatches for all 44 stations across all six main inversions. The same information is also plotted in Fig. S4,

ordered by the latitude of the station.

For most stations, the choice of the biomass burning a priori has very little influence on the final fit, as evident from the orange, green, and pink lines in Fig. 5 coinciding almost everywhere. Moreover, the a priori mixing ratios from the different inventories themselves are fairly similar. In general, a priori mixing ratios are lowest before the *GFED* inversion and highest before the *reference* inversion based on FINN2.5+VIIRS, though this does not allow for any conclusions regarding the quality




of the inventories. With all three, the a priori mixing ratios are clearly overestimated. While GFED4.1s generates the lowest a priori mixing ratios which are, therefore, closest to the observations ($\bar{J}_{\text{flask}} = 15.88$ is the smallest prior mismatch out of all experiments), this could be coincidental.

### 4.1.2 Set 2: Inversions based on different observational datasets

For the same stations as in Fig. 5, the modeled mixing ratios for the second set of experiments (*satellite only*, *station only*,
and *full satellite*) based on different observational input datasets are shown in Fig. 6. At the resolution of the model employed in this study, even within the zooming region (up to $3° \times 2°$), only minor differences in a posteriori mixing ratios are found between the *full satellite* inversion (green lines) versus the *reference* inversion (orange lines), i.e. for the sake of this study, those datasets are equivalent. This equivalence is also emphasized by very similar mismatch values in Table 2. In the *station only* inversion, where the satellite observations are excluded altogether (brown lines), the fit to the flask measurements gets
slightly better (lowest $\bar{J}_{\text{flask}}$ in Table 2), though changes are mostly minimal. Larger changes are found when comparing the former three inversions to the *satellite only* inversion (pink lines), in which the model is not driven by the flasks at all. In Table 2, this leads to a significantly larger $\bar{J}_{\text{flask}}$, compared to all the other experiments, yet the mismatch is still lower than for the a priori. This shows that the error inflation factors introduced in Sect. 3.2.2 have been chosen to meaningful values, because the station fits do not significantly degrade due to the satellite observations in the combined inversions.

Stations at high ($> 55°$) northern latitudes, like Alert in Fig. 6a, exhibit a poor fit quality for the *satellite only* inversion. During northern hemispheric summer, mixing ratios stay close to the a priori and much higher than the flasks, while in northern hemispheric winter they fall too low, diverging from the a priori and the flasks. This implies that these stations systematically have large mismatches. To illustrate that the fit at other stations is better, we calculated $\bar{J}_{\text{flask}}$ only for stations south of $55° \text{N}$ in the third and fourth row of Table 2. While $\bar{J}_{\text{flask}}$ is significantly reduced for the *satellite only* inversion, it stays almost constant
for all other experiments. This implies that the satellite observations specifically are insufficient to constrain these stations at high northern latitudes, while the model itself is well capable of capturing them. In the *satellite only* inversion, during northern hemispheric wintertime, there are very few observations in this region, due to little light and high cloud coverage. Therefore, the divergence from the a priori is likely driven by an unbalanced budget in the northern tropical and subtropical regions, where emissions all year round are heavily reduced as shown in Sect. 4.3 below. It is cheaper for the model, in terms of the cost
function, to diffuse the decrements over a larger area and shift a part of them to higher northern latitudes, than to have even deeper localized decrements in the tropics.

Aside from the northern stations, there are a few other stations that are problematic for the model to capture. The most extreme example of these issues is the station in the Assekrem (ASK) shown in Fig. 6b, where the satellite drives the model to much lower mixing ratios than the flasks. This underestimation can be clearly seen by the very low a posteriori mixing ratios for
the *satellite only* inversion (pink line), and by the *reference* inversion (orange line) ending up consistently lower than the *station only* inversion (brown line), which is seldom the case for other stations. For this specific station, this effect is likely amplified by its positioning within the Sahara desert, where satellite observations are plentiful due to high albedo and little cloud cover, but might also be adversely affected by dust. This oversampling causes the satellite observations to gain a relatively large





**Figure 6.** Modeled a priori (dotted line) and a posteriori (solid lines) mixing ratios sampled at the locations of the stations as well as the flask observations (blue crosses) for six example stations and four inversions with different observational datasets. For each observation, the corresponding measurement error is indicated as well. Lines are color-coded based on the observations used: The orange lines represent the *reference* inversion and are identical to the orange lines from Fig. 5. In green the *full satellite* inversion is shown, which also uses a combination of satellite and flask observations. The pink and brown lines represent the *satellite only* and *station only* inversions, respectively. Note that because all inversions are based on the same a priori emissions, the single dotted black line holds for all four inversions.





weight in the cost function compared to the flasks at that location, causing the *reference* inversion to slightly diverge from the
flask observations. Assekrem is also a high-altitude site, which could potentially be problematic with the limited representation
of topography in the model. When considering the resulting emission increments (Sect. 4.3) it appears that the model is not
capable of capturing this station properly. Another problematic station is Hohenpeissenberg (HPB), shown in Fig. 6c, where
the *satellite only* inversion, again, suggests much lower mixing ratios. Note the larger range on the vertical axis. Similar,
albeit less pronounced results are found for Ochsenkopf station (OXK), which is relatively close to Hohenpeissenberg station
geographically. Both are located on mountains at high altitudes. Therefore, as mentioned earlier, the coarse resolution of the
model and its limited representation of topography might adversely affect the results there. This misrepresentation will also be
further discussed in Sect. 4.3 below, where these specific stations are found to lead to unrealistically high emission increments,
similar to Assekrem station. As for the stations at high northern latitudes, these three stations (ASK, HPB, and OXK) degrade
the global mean error-weighted mismatch exceptionally strongly. To illustrate this, in the fifth and sixth row of Table 2 we
calculated $\bar{J}_{\mathrm{flask}}$ for all but these stations. Again, $\bar{J}_{\mathrm{flask}}$ for the *satellite only* is reduced strongly. However, there are also slight
decreases for the other experiments, suggesting that the model overall has an issue with properly representing these stations.

Nonetheless, most other stations, regardless of geographical location, show good fits for all four investigated combinations
of observational input. As examples for northern tropics, high southern latitudes, and southern tropics, Mauna Loa, Palmer
Station, and Rapa Nui, respectively, are shown in Figs. 6d-f. Most notably, the *satellite only* inversion manages to closely
follow the flask measurements, despite them being not assimilated. This can be seen in the seventh and eighth row of Table
2, where both, the stations north of $55°$ N and the problematic stations (ASK, HPB, OXK) are excluded from the calculation
and $\bar{J}_{\mathrm{flask}}$ for the *satellite only* inversion gets much closer to the other experiments. These good fits suggest that inversions of
current events driven solely by TROPOMI observations are feasible, as long as the region of interest is well south of around
$55°$ N.

## 4.2 Mixing ratio mismatch to the satellite observations

In the final two rows of Table 2, we calculated the total error-weighted mismatch $J_{\mathrm{sat}}$ between satellite observations and model
for all main inversions, as

$$J_{\mathrm{sat}}(\boldsymbol{x}) = \sum_i \left[ \frac{(y_{\mathrm{sat},i} - \mathbf{F}(\boldsymbol{x})_i)^2}{\varepsilon_{\mathrm{O},i}^2} \right], \tag{12}$$

where $y_{\mathrm{sat},i}$ are the satellite observations with observational errors $\varepsilon_{\mathrm{O},i}$, and $\mathbf{F}(\boldsymbol{x})_i$ is the model sampled at that measurement,
with the averaging kernel applied. Supplemental Figure S5 shows the temporal (monthly) and spatial ($12° \times 12°$ grid) dis-
tribution of the total error-weighted mismatches for all main inversions. Unlike for the mean error-weighted mismatch $\bar{J}_{\mathrm{flask}}$
between the flasks and the model introduced in the previous section, we did not divide by the number of observations here,
hence we calculated the total instead of the mean mismatch. Considering the total mismatch was necessary, because the number
of observations in the *full satellite* inversion is much larger than in all other inversions that use the gridded super-observations.
Therefore, the mean error-weighted mismatch for the non-gridded observations is much smaller, i.e. each single observation
bears a smaller weight in the inversion. By design, the super-observations have smaller error than each single observation they





are made up from (Sect. 3.2.1) and the error of satellite observations in the *full satellite* inversion is inflated the strongest (Sect. 3.2.2). Overall, the total mismatch leads to comparable numbers, in this case, while the mean mismatch would not. Again, as for the stations in the previous sections, more detailed data can be found in the supplement, where Figs. S6 and S7 show the latitudinal distribution of the mean a priori and a posteriori mismatch between the model and the satellite observation in 12° bands for all six main inversions.

Generally, the results are similar to the ones for the stations above. When considering the first set of inversions (*reference*, *noVIIRS*, *GFED*), the a priori mismatch is again smallest for *GFED* and largest for *reference*, and for the a posteriori mismatch this is inverted again. For the second set, the *satellite only* inversion results in the best fit to the satellite observations, while the *station only* inversion results in the worst. This is akin to the results from the previous section, where the *station only* inversion had the best fit to the station data and the *satellite only* inversion had the worst fit. As outlined above, the mismatch for the *full satellite* inversion is special, because it is calculated with respect to the non-gridded dataset. Regardless, the mismatch reduction is comparable to the *reference* inversion.

The mismatches mainly originate from regions known for biomass burning, such as central and southern Africa, northern South America, eastern North America, Indonesia, and Siberia. Even the $0.5° \times 0.5°$ grid of the super-observations is fine compared to the model resolution of $3° \times 2°$ or $6° \times 4°$. Therefore, any biomass burning event that leads to steep gradients in the observations cannot be resolved in the model and will lead to mismatches between the modeled and observed mixing ratios.

The global a posteriori mismatches also vary in time and are largest in August during the height of the burning season. More details on this can be found in Supplemental Figs. S8 and S9, which show the global total prior and posterior mismatch between the satellite observations and the model for each month of each of the main inversions. This spike in August is especially pronounced in the *station only* inversion, where the mismatches already rise in July and slowly taper off over the following months. For this inversion, in addition to the coarse model resolution, the station measurements are too sparse in time and space to properly capture individual biomass burning events and only constrain the increases in the resulting well-mixed background mixing ratios. Similar as for the stations, the a priori mismatches are initially low in June and steeply rise over the following three months. The good initial fit shows that the *spin-up* inversion manages to properly harmonize the modeled mixing ratio with the observations, as intended. The following rise in mismatches also illustrates the suspected unbalanced budget that causes CO to accumulate in the model.

Supplemental Fig. S10 provides a closer look at the monthly lateral distribution of the total a posteriori mismatch between the satellite observations and the model for each inversion compared to the *reference* inversion, i.e. when and where each inversion preformed better or worse than the *reference* inversion. For the first set of inversions, it becomes apparent that, while the *GFED* inversion leads to worse mismatches overall, the mismatches in Indonesia are slightly smaller compared to the *reference* inversion. Additionally, *noVIIRS* and *GFED* perform slightly better than *reference* in central Africa in the beginning of the burning season in August to October, but the *reference* inversion performs better there for the rest of the year.

Further analysis of the second set shows that for the *satellite only* inversion the lower mismatch originates mostly from the northern hemisphere. Curiously, the mismatch towards the satellite observations around Rapa Nui in the southern Pacific is





significantly increased (by roughly 50 %) in the *satellite only* inversion for the period October to December compared to the *reference* inversion, i.e. in that region, the additional use of flask measurements in the *reference* inversion leads to a better fit to the satellite observations than using the latter on their own. This apparent contradiction can be resolved by considering that the

mixing ratios at such remote locations are, on the one hand, only weakly constrained by the sparse satellite observations over the oceans and, on the other hand, are strongly influenced by transport from distant, land-bound source regions (Daskalakis et al., 2022), which are much stronger constrained by the satellite observations. The addition of the high-confidence flask measurements from the Rapa Nui station causes the model to diverge from the a priori towards higher emissions around that station, which also better fit the (sparse) satellite observations in that region.

For the *station only* inversion, especially large mismatches are observed over northern Africa during the full inversion period. This is most likely related to the issues with the station in the Assekrem outlined in the previous section. During the burning season (July–September) the mismatches in the *station only* inversion are most pronounced over continental Asia, northern and central Africa, northern South America, eastern North America, and the oceans in between those regions. Towards the end of the year, large mismatches are also found around Indonesia. Notably, the *station only* inversion shows a degrading fit

to the satellite observations in high northern latitudes ($> 55°$ N), i.e. the a posteriori mismatch there is worse than the a priori mismatch (see also Fig. S6). This is the only place and time where a degrading fit occurs. As mentioned, all of this behavior is to be expected from the *station only* inversion, since the sparse station network cannot capture the full spatial and temporal variation of all biomass burning events globally.

While the mismatches for the *full satellite* inversion are problematic to compare directly to the other inversions due to the

much larger number of observations and the error inflation, the mismatches appear to be smaller in remote regions and larger in active biomass burning regions, compared to the *reference* inversion. This mismatch distribution is expected, because the higher resolution of the full satellite observations implies finer and more pronounced structures from the individual biomass burning events, which the model can resolve even less.

Interestingly, the mismatches from all main inversions converge in the southern hemisphere, i.e. even the *station only* inver-

sion fits the satellite observation just as good as the *reference* or even the *satellite only* inversion. This shows that not only is each dataset on its own sufficient to constrain the (remote) southern hemisphere, but they also end up at roughly the same result there.

### 4.3 Optimized global emission fields

#### 4.3.1 Secondary production

Figure 7 provides a global overview of the optimized secondary CO production from VOCs including $CH_4$ for September 2018 and a comparison to the a priori emissions for the *reference* inversion. In Panels (c) and (d) the absolute and relative differences between the a priori (Panel (a)) and a posteriori (Panel (b)) are shown. For comparison, the relative emission increments for the *noVIIRS* and *GFED* inversions can be found in Panels (e) and (f), respectively. September was arbitrarily chosen, because it is in the center of the inversion period and the results found for the other months are fairly similar. The differences that



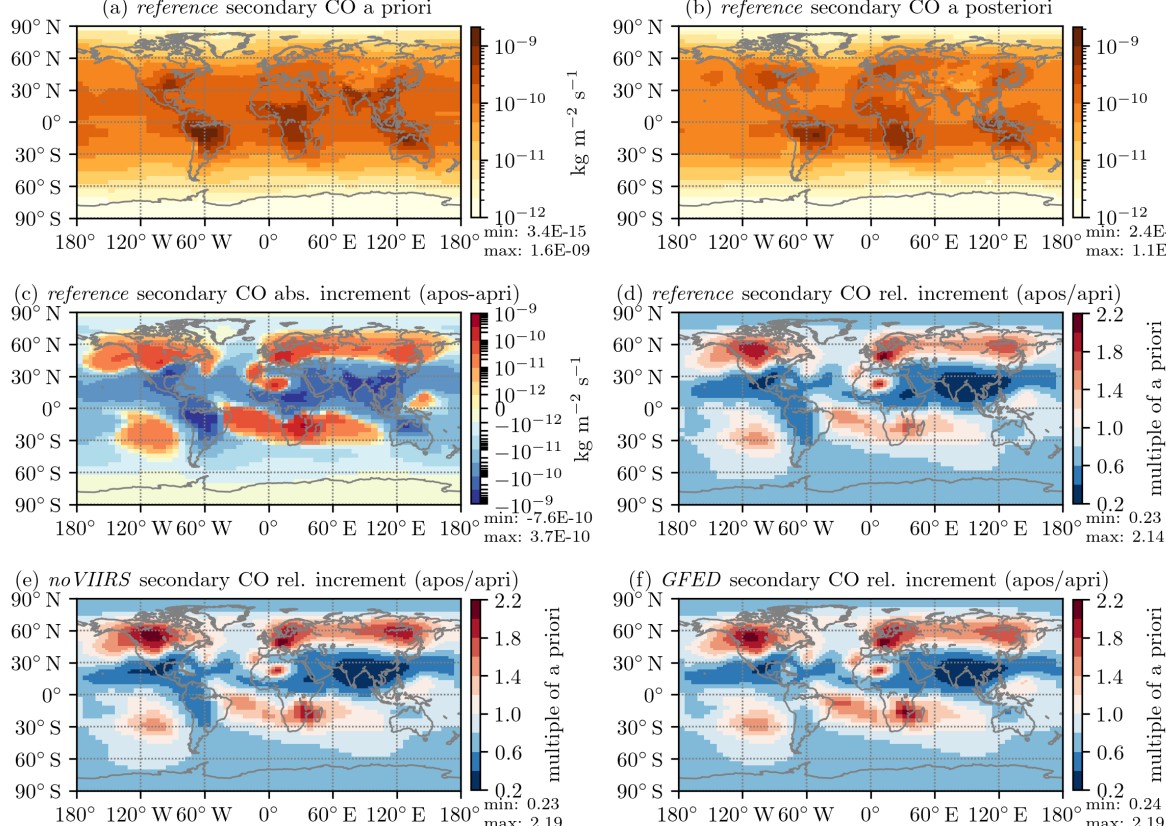

**Figure 7.** Global secondary CO production for September 2018 for the first set of experiments. The first four panels belong to the *reference* inversion (based on FINN2.5+VIIRS) and show (a) the a priori emissions, (b) the a posteriori emissions, (c) their absolute difference, and (d) the factor by which the emissions increased. Panels (e) and (f) show this factor for the *noVIIRS* and *GFED* inversions, respectively. Note the logarithmic color scales in the first three panels.

occur over time are small and limited to variations in amplitude, but not in space. This is to be expected, considering the strict temporal correlation times and spatial correlation lengths introduced in Sect. 2.3.1. Supplementary Figs. S11–S13 provide a brief overview of the relative secondary CO increments resulting from the *reference* inversion for the remaining six months of the main inversion period and comparisons of those increments to the ones shown in Fig. 7.

All main inversions result in large decrements in a band roughly between the Equator and 40° N. These decrements are 710 especially deep over China and India, as can be seen in the relative increments in Figs. 7d–f. In the later months of the inversion period, this region of large decrements shifts eastwards towards China for all experiments. This northern tropical decrement will be analyzed in more detail later on in Sect. 4.3.2, in the context of anthropogenic emission increments.

The band of decrements is accompanied by increased emissions north of 40° N, especially over central Europe, North America, and Siberia. Additional positive increments can be found between the Equator and 40° S, over the oceans, and in



southern Africa. These appear to occur in biomass burning outflow regions, and could point to a systematic error in the lifetime of CO in the model. Due to the band-like structure of the positive and negative increments, this error is possibly caused by inaccurate OH values. Further evidence for such issues with OH values can be found in Myriokefalitakis et al. (2020), where they compare their online calculated OH to the climatological fields from Spivakovsky et al. (2000) used here and find significant differences in those regions. Notably, in the full chemistry simulation, higher OH concentrations not only imply

higher CO loss rates, but also higher secondary CO production. Here, we use those production rates paired with loss rates base on the climatological OH, as pointed out in Sect. 2.3.3. Since in our inversions the loss rates are fixed, the model can only compensate for this mismatch by, in some places considerably, changing the secondary CO source.

Overall, the a posteriori secondary CO source is lower than the a priori production flux in all experiments, as can be seen in the global budgets provided in Table 3, where the posterior masses at the end of the inversion period (final masses) are

consistently lower than the prior final masses. All fluxes have been extrapolated to annual budget terms in Tg CO yr$^{-1}$, which might be misleading, because the inversion period of the main inversions includes the biomass burning season, but excludes the increased anthropogenic emissions due to heating during most of the northern hemispheric winter. Regardless, our extrapolated annual a posteriori budget terms are much closer to the ones found in literature (e.g. Zheng et al., 2019) than the a priori terms, implying that the a posteriori terms are more realistic. However, as expected, the partitioning of our emission terms is slightly

different compared to Zheng et al. (2019), with lower anthropogenic/fossil fuel CO emissions, but higher secondary CO production.

As for the stations in Sect. 4.1, the differences in the emission increments between the inversions in the first set (different biomass burning a priori) are rather small. The most striking differences are the much larger increments (up to 60 % higher final emissions) over southern Africa in both the *GFED* and *noVIIRS* inversions (Figs. 7e and 7f). These are likely related to

a known underestimation of African CO emissions in GFED4.1s as described in Nguyen and Wooster (2020) and references therein. Due to its improved small fire handling, FINN2.5+VIIRS, as used in the *reference* inversion, appears to be more capable of capturing those fires. More subtle differences are found in South America, where the *GFED* inversion only leads to minor corrections (relative increments close to 1), while the *reference* and *noVIIRS* inversions show clear decrements (final emissions reduced by up to 50 %). These decrements could be coincidental, considering the importance of OH-chemistry and

secondary CO production in that region. In the northern hemisphere, *noVIIRS* (Fig. 7e) and *GFED* (Fig. 7f) feature slightly higher increments over eastern Europe (*noVIIRS* < 10 %, *GFED* up to 35 %), North America (*noVIIRS* < 10 %, *GFED* < 20 %), and Siberia (*noVIIRS* < 15 %, *GFED* < 5 %) compared to the *reference* inversion. These differences could point to aliasing of the secondary production emission category to the biomass burning category. FINN2.5+VIIRS, which is used as biomass burning a priori in the *reference* inversion, has generally the highest emissions, mostly due to capturing small fires, which are

common in these regions. For the other two, the model attempts to capture these missing sources, in part, through increasing the emissions in the other categories. Again, this misattribution can also be seen in the budgets in Table 3, where the posterior total emitted mass is very similar for all experiments of the first set, but the distribution over the three emission categories varies considerably.





**Table 3.** Global prior and posterior budgets for all inversions, as a sum over the global and the zooming regions. The zooming column combines masses going into and coming from the communication cells between the zooming regions. For the main inversions, the $3° \times 2°$ region perceives this as a net loss through advection into these cells, while the global region perceives it as a net gain through emissions within the cells. Only the net effect is shown here. Note that the annual rates (Tg $CO$ yr$^{-1}$) are extrapolated from the emissions during the respective inversion periods, January to June (6 months) for the spin-up inversion and June to December (7 months) for the main inversions.

| experiment | | masses in Tg CO | | losses in Tg CO yr$^{-1}$ | | zooming in | emitted in Tg CO yr$^{-1}$ | | | |
| --- | --- | --- | --- | --- | --- | --- | --- | --- | --- | --- |
| | | initial | final | chemical | deposition | Tg CO yr$^{-1}$ | total | secondary | biomass | fossil fuel |
| reference | prior | 556 | 739 | -2995 | -216 | 113 | 3411 | 2179 | 613 | 618 |
| | posterior | | 584 | -2487 | -187 | 21 | 2701 | 1637 | 543 | 520 |
| noVIIRS | prior | 556 | 722 | -2904 | -206 | 102 | 3291 | 2179 | 493 | 618 |
| | posterior | | 584 | -2486 | -185 | 20 | 2699 | 1701 | 472 | 525 |
| GFED | prior | 556 | 699 | -2816 | -199 | 88 | 3172 | 2179 | 374 | 618 |
| | posterior | | 584 | -2480 | -185 | 20 | 2692 | 1766 | 366 | 560 |
| satellite only | prior | 556 | 739 | -2995 | -216 | 113 | 3411 | 2179 | 613 | 618 |
| | posterior | | 579 | -2477 | -184 | 16 | 2684 | 1627 | 545 | 513 |
| station only | prior | 556 | 739 | -2995 | -216 | 113 | 3411 | 2179 | 613 | 618 |
| | posterior | | 593 | -2580 | -192 | 23 | 2811 | 1704 | 599 | 508 |
| full satellite | prior | 556 | 739 | -2995 | -216 | 113 | 3411 | 2179 | 613 | 618 |
| | posterior | | 587 | -2501 | -188 | 22 | 2719 | 1650 | 552 | 517 |
| spin-up | prior | 646 | 670 | -2925 | -212 | 26 | 3159 | 1991 | 532 | 637 |
| | posterior | | 521 | -2349 | -181 | -77 | 2355 | 1347 | 382 | 626 |



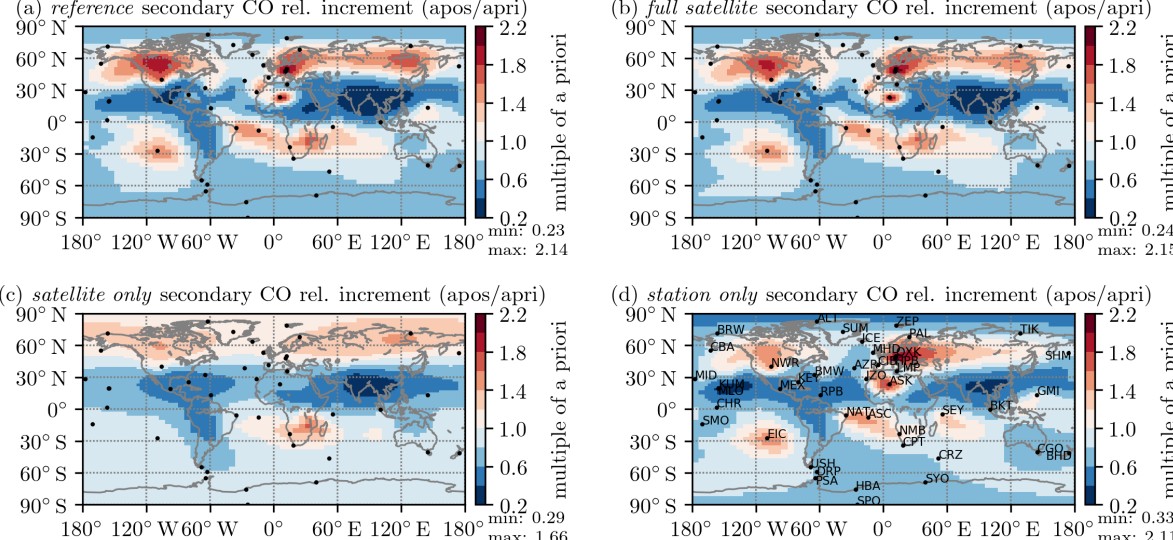

**Figure 8.** Global secondary CO relative emission increment for September 2018 for the second set of inversions, based on different observational datasets. The panels show the factor by which the emissions increased for (a) the *reference* inversion, (b) the *full satellite* inversion, (c) the *satellite only* inversion, and (d) the *station only* inversion. The locations of the surface stations are indicated with dots for easier orientation, in the last panel additionally with their station code. Note that Panel (a) of this figure is the same as Fig. 7d.

In Fig. 8, one month of the relative increments for the CO production from VOCs and CH$_4$ are shown for the second set
of inversions. Figure 8a is from the *reference* inversion based on a combination of gridded satellite observations and surface flasks. As such, the content of Fig. 7d above is repeated there for ease of comparison. Very similar results (Fig. 8b) are obtained with the *full satellite* inversion, as already shown at the surface stations in Sect. 4.1. Minor differences are visible over North America and Siberia, likely due to less aliasing to the biomass burning category. When the higher resolution observations are used, the short term and local biomass burning events are more distinct, which makes it easier for the model to capture them in
the appropriate category.

For the *satellite only* inversion (Fig. 8c) many regional features are much less pronounced. However, the broader distribution of emission increments remains the same: There are still negative increments in a band between the Equator and 40° N and over South America, and positive increments over southern Africa and the adjacent oceans. The positive increments over North America, Europe, and Siberia are weaker and appear to be spread out over the whole northern hemisphere north of around
45° N, including over the oceans. These weaker features are likely linked to the different spatial distributions of observations in the two datasets; while there are many maritime stations and stations in the remote northern hemisphere, satellite observations there are more sparse and mostly found in continental regions. Additionally, towards the end of the year, i.e. the second half of the main inversion period, there are no more satellite observations at high northern latitudes, as exemplified in Fig. 1 for





one day in early November. All of this, in combination with the spatial correlations given to the optimizer, causes the model to
prefer smooth, broad patterns to fill in any gaps.

These differences in information content between the two observational datasets stress the importance of the error inflation
(Sect. 3.2.2). If the error on the satellite observations is not inflated, the optimized emissions end up very close to the ones
from the *satellite only* inversion, because the signal from the sparse flask measurements is overshadowed. However, the current
inflation may be too large, which causes the optimizer to "overfit" certain stations that are not well captured by the model. As
can be seen in Fig. 8d for the *station only* inversion, some stations clearly drive the model away from these broad patterns and
towards strong positive regional increments. This overestimation is especially apparent for Assekrem (ASK) and Izana (IZO)
stations, which lead to large increments over north-west Africa, and Hohenpeissenberg (HPB) and Ochsenkopf (OXK) stations,
which drive emissions over central Europe up strongly. Neither of these increments are observed or supported by the satellite
observations. Notably, all of these stations are at high altitudes, potentially pointing to short-comings in the representation
of topography in the model. However, there are mountainous stations, like Mauna Loa (MLO), that are captured well by the
model.

Less pronounced examples of overfitted stations are Rapa Nui (EIC) and Tutuila (SMO), which cause positive and negative
increments over the southern Pacific, respectively. However, it should be noted that for the satellite the number of observations
over oceans to constrain those emissions is very limited and, as shown for Rapa Nui in Fig. 6f, the *satellite only* inversion still
manages to fit these stations reasonably well.

Another factor that could play a role in the context of overfitted stations is the strength of the vertical transport in TM5,
which Krol et al. (2018) find to be somewhat faster than in other models. This implies low vertical gradients in the troposphere
and that modeled tracer mass might be transported upwards before the model can be sampled at the location of the station
for comparison to the real observations. This is especially problematic for remote stations with limited surface sources in the
vicinity, such as Rapa Nui (EIC) in the south-eastern Pacific. There, the model is forced to introduce unrealistic increments
to the secondary CO source in the middle of the Pacific. Furthermore, due to the way those emissions are handled within
the model, this will introduce additional CO over the whole column (and not only at the surface), which then hampers the
comparison to the satellite observations. Similarly, for the station in the Assekrem, in the inversions that include station data,
the low vertical gradients cause the optimizer to introduce unrealistically high secondary CO emissions over the Sahara. In
contrast, those increments do not occur in the *satellite only* inversion, because the satellite observes the total column with a
very limited vertical resolution and is, therefore, less affected by the vertical gradient in the model.

Finally, even in the *station only* inversion (Fig. 8d), some station driven features appear weaker compared to the *reference*
inversion (Fig. 8a). For example, the positive increments over North America are much weaker and the spikes around the
Assekrem and in central Europe are more spread out. These weaker features are again caused by a combination of the prescribed
spatial correlations and the distribution of the available observations. While in the *station only* inversion the model prefers
broader patterns to follow the prescribed spatial correlation of the emissions, in the *reference* inversion there are satellite
observations all around the landlocked stations, which drive the model towards lower increments. Overall, the *station only*





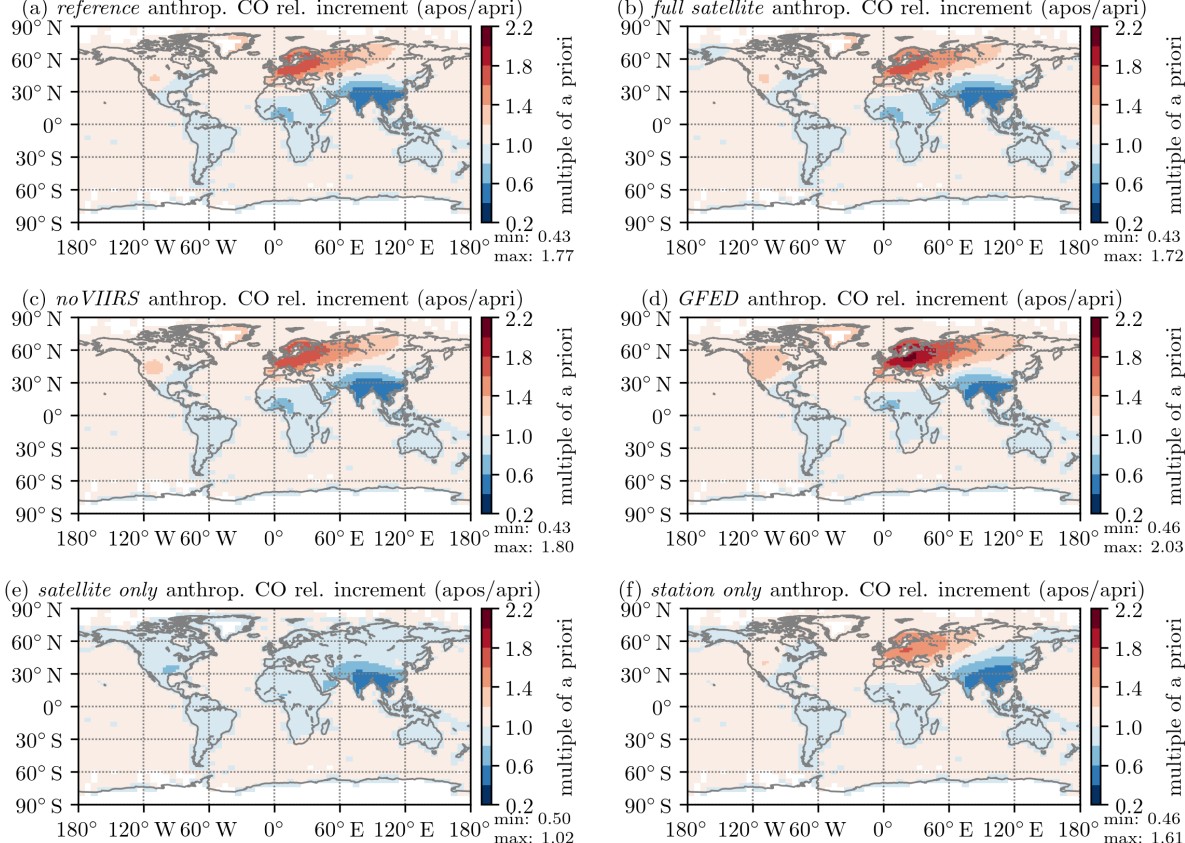

**Figure 9.** Relative global anthropogenic CO emission increments for September 2018 for all six inversion experiments. Panel (a) shows the *reference* inversion with FINN2.5+VIIRS as biomass burning a priori, and gridded satellite observations and surface flasks as observational input. The variations are (b) *full satellite* observations instead of gridded, (c) *noVIIRS* with FINN2.5 as biomass burning a priori, (d) *GFED* with GEFED4.1s as biomass burning a priori, and (e) *satellite only* and (f) *station only* to drive the inversion.

inversion is driven to the largest emitted mass of all experiments as shown in the budgets in Table 3. This is in line with the increased emissions around surface stations postulated in the context of the (too) fast vertical transport in TM5 above.

### 4.3.2 Anthropogenic emissions

To better identify the aliasing between the emission categories, Fig. 9 provides an overview of the relative increments in the optimized anthropogenic emissions for all six inversions. In Fig. 9a the relative emission increments are shown for the *reference* inversion based on FINN2.5+VIIRS, and a combination of gridded satellite observations and surface flasks. The largest changes are positive increments over Europe, and negative increments over China and India. To investigate these increments further, we must first consider that the anthropogenic a priori emissions taken from CMIP6 are projections for 2018, rather than



historical data. For China, these projections predict relatively constant emissions. However, China managed to significantly reduce its CO emissions in recent years (Kanaya et al., 2020) in the scope of air quality policies, like the Coal to Gas policy only implemented in 2013 (Liu et al., 2020). Additionally, the effect of most of these policies was somewhat offset by strong biomass burning years up until 2015 (Zhang et al., 2020), making their effect harder to assess in advance. Regardless, reduced

CO concentrations have been now observed all over China, both at surface stations (Liu et al., 2019; Zhai et al., 2019; Li et al., 2020) and from satellites (Zhang et al., 2020). This observed reduction has been linked to a decrease in emissions as calculated using inverse modeling (Zheng et al., 2018). The reduced emissions are most likely due to anthropogenic rather than natural factors (Kang et al., 2019). By 2018, the year of this study, all of this adds up to at least part of the significant offset in CO emissions that we observed.

Unlike for China, there is no clear explanation for the negative increments over India. These might be an artifact due to spatial correlation, where India's proximity to China implies that it is cheaper in terms of the cost function to reduce emissions over a larger region, rather than strongly reducing only China's emissions. This could be compounded by low observational coverage, especially with regard to surface stations, and an OH climatology not appropriate for recent years.

When compared to the *full satellite* inversion shown in Fig. 9b, again, the increments are almost the same, further justifying

the usage of gridded satellite observations on a global scale to reduce the computational cost.

The *noVIIRS* (Fig. 9c) and *GFED* (Fig. 9d) inversions are slightly worse at capturing the small fires in Europe and North America compared to the *reference* inversion. The missing small fires lead to apparent anthropogenic increments, especially for *GFED*, over Europe and western Russia to close the CO budget. Further evidence for this aliasing is provided in Table 3, where the total a posteriori emissions for the inversions of the first set are almost identical, but the partitioning over the emission

categories differs significantly. As such, *GFED* has around 33 % lower biomass burning emissions compared to *reference*, but almost 8 % higher both secondary production and anthropogenic emissions.

For the *satellite only* inversion, the relative anthropogenic emission increments are pictured in Fig. 9e. They stay relatively close to, but below, 1 globally, i.e. the inversion mostly agrees with the a priori. Over India and China, again, a clear decrement is visible. Notably, there is no increment over Europe, in contrast to what we find when flask observations are included. In Sect.

4.1, this smaller increment caused the station at Hohenpeissenberg (Fig. 6c) to be considerably underestimated in the *satellite only* inversion.

The *station only* inversion shown in Fig. 9f leads to very similar results in terms of anthropogenic increments compared to the *reference* inversion. This shows how well the NOAA station network on its own is capable of constraining the global broad-scale background emission patterns. Differences include smaller increments over Europe and smaller decrements over

Africa and an apparent shift of the decrement over India and China towards the East. The latter may be explained by a lack of background stations and, therefore, a lack of observations in that region, causing the decrement to be smoothed out due to spatial correlation.

Overall, the anthropogenic increments shown in Fig. 9 compared to the ones for the secondary CO production in Figs. 7 and 8 show similar general structures, with decrements in China and India and increments in Europe. However, there are noticeable

differences both in finer scale spatial details, for example, the anthropogenic increments over Europe are more spread out





towards Eastern Europe, and large scale patterns, with much smaller relative increments for North America. Generally, the ratios of a priori to a posteriori emissions, i.e., the relative emission increments, are not the same for all three categories. In other words, while there is some aliasing, the inversion setup is still capable of simultaneously optimizing multiple emission categories, which is ensured in the following ways:

Firstly, because of the different a priori errors, even in regions with similar spatial structures, the amplitudes of the relative emission increments differ significantly. Secondly, the different correlation lengths and times for each emission category, as introduced in Sect. 2.3.1, ensure that only the biomass burning category is capable of capturing short and local events. Conversely, long-lasting, large-scale mismatches could still lead to aliasing across all categories, as is the case, for example, over China. Thirdly, the a priori emissions of all three categories feature different spatial structures. These a priori structures,

combined with enforcing spatial and temporal correlation, imply that it is cheapest for the model to change emissions following the 'spatial signature' of the correct source category, rather than evenly distributing the increments over all categories. An example for this can be found over North America, where the anthropogenic emissions are barely changed, while there are significant changes in the secondary CO production.

### 4.3.3 Biomass burning

An in-depth analysis of the optimized biomass burning emissions is not included in this study, because the low model resolution is not sufficient to capture individual burning events. This promotes aliasing between the emission categories, where the biomass burning emissions are in- or decreased in large regions co-located to the patterns observed in the secondary CO production. As an example of this, Fig. S3 shows the absolute biomass burning increments for 15 September 2018, the day in the center of the period analyzed above. Because the temporal variability in the secondary CO production is low, the biomass

burning emissions also remain relatively constant in time.

### 5 Conclusions

We introduced TROPOMI satellite observations into the TM5-4DVAR inverse modeling suit to optimize global CO emissions from three distinct emission categories (biomass burning, anthropogenic, and secondary production) in a set of six inversion experiments. The model ran at a relatively coarse resolution of up to $3° \times 2°$, which allowed for the use of satellite super-

observations gridded to $0.5° \times 0.5°$ to reduce the computational cost. Compared to the inversion based on the full-resolution (up to $7 \times 7 \, \mathrm{km}^2$) satellite observations, differences in the final mixing ratios and optimized emission fields were minimal. Yet, the computation time per iteration was around 25 % longer for the full resolution inversion. However, at $3° \times 2°$ resolution, the model could not properly resolve the spatial scale of individual biomass burning events. This resulted in heavy aliasing of the biomass burning emissions to the other emission categories. In future studies, using additional observations to further constrain

emissions from specific sources or by employing a finer zooming region could improve model performance. With the latter, such an inversion could make use of the full potential of the TROPOMI observations.



The comparison of model results and observations is vastly improved by the inversion and the a posteriori mixing ratios closely follow the observed values. Notably, this even holds true in regions like China and the North Pacific, where the a priori strongly overestimated the mixing ratio and very large emission decrements are required to reach a good a posteriori fit.

The overestimated a priori mixing ratios in those regions reveal inconsistencies between the OH climatology used to simulate chemical loss, and the secondary CO production terms taken from the TM5-MP model. This will be further investigated in a study currently in preparation. For the inversion based only on satellite observations, sizable mismatches between model results and flask measurements remain for stations at high northern latitudes. These mismatches can be explained by considering that mixing ratios at high northern latitudes, on the one hand, are poorly constrained by the satellite observations, especially towards

the end of the year, and, on the other hand, are governed by transport from the (well-constrained) mid latitudes, which leaves little leeway for the optimizer. Additionally, in the inversions based on flask measurements, there are very large increments around high-altitude stations. These increments are most likely linked to the coarse model orography that comes with the overall coarse model resolution. Despite good coverage in those regions, the inversion based only on satellite observations neither confirms nor reproduces those strong increments. As such, for future inversions in this framework, an increased model

representation error should be applied to those specific stations, to avoid biasing results by overfitting.

In the southern hemisphere, we find very similar results across all inversions, regardless of the observational dataset(s) (satellite, stations, or both) used. This indicates that, in the southern hemisphere, either dataset is equally capable of and sufficient for constraining the background emissions and leads to the same mixing ratios. Potentially, these promising results could allow for inversions based solely on TROPOMI observations, so long as the region of interest is sufficiently far south of

$55°$ N. There, as well as for validation, bias correction, and overall confidence in the optimized emissions, the surface flasks still play a crucial role in the inversion. By using the TROPOMI observations on their own, the long analysis cycle of the surface flasks could be circumvented and specific events could be investigated using this model in a more timely manner (within weeks rather than months), and only be verified against and adjusted by the flasks at a later stage.

Overall, the most reliable results are found from inversions using both datasets, because they complement each other in

multiple ways. Firstly, their spatial coverage differs slightly – while the satellite observations are mostly valid over land but sparse over the oceans, most background stations are located on remote islands or in coastal settings. Secondly, both datasets on their own have very limited information on the vertical tracer distribution, where the flasks probe only the surface layer and the satellite observations provide only total column mixing ratios. Combining those datasets can yield better constraints on the vertical tracer distribution in places where in situ and satellite observations are co-located. Finally, in a joint inversion, the

satellite observations are implicitly verified versus the flask measurements and it becomes possible to identify potential biases in the satellite observations. However, when using both datasets at once, the technical limitations of both apply, i.e. the high computational cost from using the satellite observations, and the long analysis cycle of the flask measurements.

*Code and data availability.* A snapshot of the full TM5-4DVAR model source code and the rc-files (settings) used for all inversions presented are available at Nüß et al. (2024a). Our implementation of the gridding approach to obtain the $0.5° \times 0.5°$ TROPOMI super-observation is



available at Nüß et al. (2024c). All other analysis and plotting scripts used throughout this manuscript as well as any relevant model in- and outputs are collected and available at Nüß et al. (2024b).

*Author contributions.*  Conceptualization, J.R.N., M.V., M.C.K. and N.D.; methodology, J.R.N., N.D. and M.C.K.; software, J.R.N., F.G.P.; formal analysis, J.R.N.; investigation, J.R.N.; resources, M.V.; data curation, A.G. and O.S.; writing—original draft preparation, J.R.N.; writing—review and editing, N.D., M.V., M.C.K., M.K., M.B., O.S., A.G. and F.G.P.; visualization, J.R.N.; supervision, N.D., M.V. and
M.C.K.; project administration, M.V., M.K. and M.B.; funding acquisition, M.V., M.K. and M.B. All authors have read and agreed to the published version of the manuscript.

*Competing interests.*  The authors declare that they have no conflict of interest.

*Acknowledgements.*  The simulations were performed on the HPC cluster Aether at the University of Bremen, financed by DFG within the scope of the Excellence Initiative. We further acknowledge financial support from the University of Bremen.

Part of this research is funded by the Deutsche Forschungsgemeinschaft (DFG, German Research Foundation) under Germany's Excellence Strategy (University Allowance, EXC 2077, University of Bremen). We gratefully acknowledge the support received from the "U Bremen Excellence Chair Program".

This publication contains modified Copernicus Sentinel data (2018). Sentinel-5 Precursor is an ESA mission implemented on behalf of the European Commission. The TROPOMI payload is a joint development by the ESA and the Netherlands Space Office (NSO). The Sentinel-5
Precursor ground-segment development has been funded by the ESA and with national contributions from the Netherlands, Germany, and Belgium. The TROPOMI/WFMD retrievals used here were performed on HPC facilities of the IUP, University of Bremen, funded under DFG/FUGG grants INST 144/379-1 and INST 144/493-1.

This research also received funding from the European Space Agency (ESA) Climate Change Inititative (CCI) via project GHG-CCI+ (ESA contract No. 4000126450/19/I-NB).

The TM5-MP simulations were done in scope of the FORCeS project, funded by the European Union's Horizon 2020 research and innovation programme under grant agreement No. 821205.

We acknowledge the World Climate Research Programme, which, through its Working Group on Coupled Modelling, coordinated and promoted CMIP6. We thank the climate modeling groups for producing and making available their model output, the Earth System Grid Federation (ESGF) for archiving the data and providing access, and the multiple funding agencies who support CMIP6 and ESGF.



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
