# Peer review of "Top-down CO emission estimates using TROPOMI CO data in the TM5-4DVAR (r1258) inverse modeling suit"

_EGUsphere, 2024_

## Author Comment (AC1)

We would like to thank reviewer 1 for the effort, interesting questions, and constructive comments. Below we address the points one by one. The reviewer's comments are listed in cursive, with our answers in blue and excerpts from the revised text in red.

*Overall – This paper provides a useful study of trade-offs for the inversion of carbon monoxide emissions for both the data and a priori considerations. I find that the claim that the method could be suitable for near-real-time inversions is not supported by the results. However, the main conclusions are sound and I think the paper could be published after addressing some concerns.*

*1. Near real-time suitability: No timing trade-offs were presented, in fact the only discussion of this was that 5 real-world days were required for each inversion and more for satellite full res. Also, the grid resolution did not allow characterization of biomass burning events, a primary motivation for near-real-time. It was suggested regional analyses, with a finer zoomed resolution, would be possible, but these were not demonstrated here.*

We thank the reviewer for the comment. Discussions about creating a near-real-time setup as a longer-term goal were largely removed in a previous iteration of the manuscript. We apologize for any confusion caused by the remaining references, which have now been removed.

*2. Use of OH monthly climatological fields (L. 137) should have more discussion of why this choice is applicable for the TROPOMI time range.*

OH in the global atmosphere is relatively well buffered, and the Spivakovsky climatology still complies with observed methyl chloroform loss rates. This climatology is still used in studies investigating recent periods. More appropriate OH fields are being explored in ongoing investigations. To address the raised concern, a small paragraph (L153ff) has been added to the manuscript:

"Jiang et al. (2017) show that OH is well buffered in the atmosphere on a global scale over the past decades, as indicated by a low month-to-month variability in the methyl chloroform loss rate, and as such the TransCom OH climatology is still considered applicable to recent years, as in e.g. Naus et al. (2022)."

*3. The only comparisons are for the surface CO flask observations. It would be of interest to see comparisons to other independent CO satellite observations.*

We thank the reviewer for this useful suggestion. However, we believe that such a comparison would extend beyond the scope of this manuscript. As an outlook, combined inversions driven by TROPOMI and IASI or MOPITT are planned for the future. At this stage, a direct comparison of the inversion results to other datasets is not trivial due to their different vertical sensitivities in combination with other model limitations. Most notably, the OH climatology needs more work before such an effort becomes meaningful.

*4. Other suggestions: L 66: should include the following reference: Naus et al. (2022), Sixteen years of MOPITT satellite data strongly constrain Amazon CO fire emissions, Atmospheric Chemistry and Physics, 22(22), 1473514750, doi:10.5194/acp-22-14735-2022.*

This is a valid suggestion. Given that the recommended paper was not published when this manuscript was first submitted, we have now included it where appropriate. Throughout the manuscript, references to recent studies have been added.

*5. Plots: Lines indicating color in the plot legend are difficult to distinguish - maybe make these thicker.*

Fig. 5 and Fig. 6 have been updated accordingly.

*6. Readability suggestions:*

We thank the reviewer for these suggestions. Corresponding updates have been incorporated throughout the manuscript. For the sake of brevity, they are not repeated here.

---

## Author Comment (AC2)

We would like to thank reviewer 2 for the effort, interesting questions, and constructive comments. Below we address the points one by one. The reviewer's comments are listed in cursive, with our answers in blue and excerpts from the revised text in red.

*Summary: This study aims to provide global CO emission estimates for late 2018, particularly in the northern hemisphere, using a top-down inverse modeling approach. By incorporating TROPOMI satellite observations into the TM5-4DVAR model and further constraining emissions with NOAA surface flask measurements, six experiments were conducted to assess the impact of different emissions and observational datasets on inversion outcomes. The main findings of this paper are: 1) The satellite-only inversion closely matches flask measurements south of 55°N, suggesting suitability for real-time applications; 2) Up to 75% emission reductions in China and India, with reductions in China attributed to policy changes; 3) Outstanding issues were identified to include underestimation of OH causing lower emissions, and localized emission increments over Europe and the Sahara.*

*Main Comments: While CO inversions have been conducted for over 20 years, there are still gaps in our understanding of CO sources and sinks despite availability of satellite data. This makes this paper potentially a relevant contribution to the scientific community and this journal given the importance of CO in understanding atmospheric composition. The authors have also described their inverse methodology in quite (and understandably) in detail which potentially can enable easier connection to relevant findings and issues identified in making the inverse framework suitable for near-real time applications. However, several concerns with this manuscript require revision:*

*1. The title "Efficacy of High-Resolution Satellite Observations" may be misleading, as approximations (e.g., model resolution and spatiotemporal scales of inversion) challenge this claim, and the findings don't fully support it.*
We agree with this assessment. The title has been revised to:
"Top-down CO emission estimates using TROPOMI CO data in the TM5-4DVAR (r1258) inverse modeling suit"

*2. While the approach shows promise in reducing uncertainties compared to flask measurements, its suitability for near real-time inversions is not adequately demonstrated in this manuscript.*
Discussions regarding a near-real-time setup as a longer-term goal had been removed in a previous iteration of the manuscript. We apologize for any confusion caused by the remaining references in the manuscript that were now removed.

*3. Verification is limited, as there's no comparison with other satellite observations or top-down emission estimates for this period. Although TROPOMI offers extensive coverage, it has limitations due to its SWIR-only focus; integrating IASI, MOPITT, and CrIS would better constrain synoptic CO patterns and widespread anthropogenic CO, which flask data alone cannot capture. One way to alleviate this are: a) comparison with other top-down estimates, b) to make objectives to be more focused.*
We thank the reviewer for this useful suggestion. As mentioned in the response to the third comment of the first reviewer (RC1), this manuscript focuses primarily on TROPOMI data and flask observations. For additional details, please see our response there (AC1). However, following the reviewer's suggestion, we have rewritten the objective to more clearly set our study apart from previous studies found in the literature and added a more in-depth comparison of the budget terms found by our inversion to the ones from other inversion studies. The objective at the end of the Introduction (L105ff) now reads:
"In this study, we investigate the added value of the new TROPOMI data for constraining global CO emissions in the TM5-4DVAR inverse modeling suit. Previous studies have already investigated the efficacy of TROPOMI observations for constraining the global atmospheric CO abundance (Inness et al., 2022) or CO emissions at regional to sub-city scales (Borsdorff et al., 2019, 2020; Sun, 2022; Tian et

al., 2022; Shahrokhi et al., 2023). Our study provides global CO emission estimates with a focus on the northern hemisphere in the second half of 2018. In addition to introducing TROPOMI observations into TM5-4DVAR, we have updated several input datasets, including the a priori emissions, and improved the methodology for handling satellite observations, most notably the weighting of multiple observational datasets in inversions, compared to previous studies using TM5-4DVAR (e.g. Krol et al., 2013; Nechita-Banda et al., 2018; Naus et al., 2022). We have divided the investigation of all of these changes into a series of experiments, in which we run the same inversion multiple times, each time with slightly different settings."

The added comparison with other top-down estimates can be found in L743ff and reads:

"Overall, the a posteriori secondary CO source is lower than the a priori production flux in all experiments, as can be seen in the global budgets provided in Table 3, where the posterior masses at the end of the inversion period (final masses) are consistently lower than the prior final masses. Naus et al. (2022), who used a similar setup, also found too high secondary CO production. All fluxes were extrapolated to annual budget terms in Tg CO yr$^{-1}$. This extrapolation may lead to misleading results when compared to budget terms published elsewhere, because the inversion period of the main inversions includes the biomass burning season, but excludes the increased anthropogenic emissions due to heating during most of the northern hemispheric winter. With this caveat in mind, we compare our prior and posterior budget terms with values from other inversion studies with different setups, namely to Jiang et al. (2017), who assimilated MOPITT CO and methyl chloroform surface measurements in the GEOS-Chem model, Müller et al. (2018), who assimilated IASI CO in the IMAGES model, and Zheng et al. (2019), who assimilated MOPITT CO in the LMDz-SACS model. A detailed comparison of these three studies can be found in Elguindi et al, (2020). Compared to the results of either of those studies, our extrapolated annual a priori budget terms for secondary CO production and chemical loss of CO to OH are far too large. However, our posterior chemical loss falls between the values found in Müller et al. (2018) and Zheng et al. (2019) and our posterior secondary CO production, while still larger, is much closer to what those studies found than our prior. This improved agreement implies that our a posteriori terms are more realistic than the a pirori ones. Note that our secondary production implicitly includes ocean and biogenic CO. While the total production and loss terms show reasonably good agreement with the aforementioned studies, the partitioning by source category of our emission terms differs slightly. Our anthropogenic/fossil fuel a posteriori CO is close to that found by Müller et al. (2018) and Jiang et al. (2017), but significantly lower than that reported by Zheng et al. (2019). In contrast, our biomass burning estimate is close to the multi-year mean of Zheng et al. (2019). However, due to the high year-to-year variability in biomass burning emissions, as shown by both Müller et al. (2018) and Zheng et al. (2019), this result is difficult to interpret, especially since neither study covers 2018."

*4. Methodology presentation could be clarified for readers less familiar with inverse modeling.*
While we tried to implement further clarifications, additional comments as to which parts are still unclear are welcome.

*5. Enhance overall clarity, particularly by: a) clearly articulating key gaps in CO inversion methodology and current top-down estimates, and b) deepening the discussion of noteworthy findings that are better supported by experiments.*
The introduction has been extended to better contextualize our manuscript within the existing body of literature, as detailed in the response to comment 3. above.

*Specific Comments:*
*1) Recommend changing the title to reflect main findings supported by experiments.*
See response to comment 1. above.

*2) While there is a good description of the methods, it may be more clear and easy to follow with an addition of a table listing all key approximations (resolution of observations and models, inflation pa-*

*rameters, super-observation parameters, inversion spatial and temporal windows, spin-up, specification of errors incl. length scales).*

The requested information has been aggregated in a new supplementary Table S1, with the inflation factors added to Table 1.

*3) Abstract: While the length constraints are understood, several statements need clarification for clarity and impact:*
*a) What is the significance of 55°N, and why would capturing measurements below this latitude suggest suitability for near real-time inversions?*

Reference to near real-time inversions has been removed, see response to comment 2. above.

*b) "Attributed to policy changes" – could you please specify which policies?*

The Chinese Coal to Gas policy has been added as an example, the sentence now reads:

"Part of the reduction in China can be attributed to policy and technology changes (e.g. Coal to Gas)."

*c) "Appears to be underestimated" – is this based on comparisons with other OH fields?*

Yes, we found large differences between the TransCom climatology used in this study and the OH fields produced by the TM5-MP model. In the manuscript, we describe an imbalanced prior budget, which is either caused by too low loss or too high production. Potentially more appropriate OH fields are the subject of further investigation in a follow-up study.

*d) The last two sentences are unclear, especially "model's limited capabilities to capture"; could you please provide more detail?*

The Abstract has been changed accordingly, with the last two sentences, due to length constraints, simplified to:

"In the experiments that include the surface flask measurements, we find strong localized emission increments over Europe and the Sahara, which are traced back to limitations of the model in reproducing point measurements on mountain tops."

*4) Line 32. Can you please rephrase? What do you mean by 'dividing them up by source categories.*

The sentence has been rephrased and a clarification was added. It now reads:

"Estimating regional CO emissions and partitioning them by source category (i.e. distinguishing CO from secondary production, fossil fuel combustion, and biomass burning) on a global scale is challenging."

*5) Line 34. 'they carry insufficient information' . Please elaborate*

The following elaboration has been added to explain that statement:

"Global remote sensing instruments usually feature very limited vertical resolution and cannot inherently distinguish when, where, and by what process (secondary production, biomass burning, etc.) each observed CO molecule was produced. In addition, the temporal resolution of global remote sensing instruments at a given location is limited to their revisit period (typically on the order of days), which may be insufficient to adequately resolve rapid events, such as biomass burning. The temporal coverage might be further reduced when clouds or other data quality issues make observations temporarily impossible."

*6) Line 35. What do you mean by 'incorporating some additional information?*

Since this was detailed in the following paragraph, the redundant mention has been removed and the sentence now reads:

"However, indirect estimation of CO emission sources from remote sensing data is possible using either bottom-up or top-down approaches."

*7) Line 37-38.'process that caused the emission is measured'; 'emissions can be extrapolated'. Please rephrase to make it more accurate.*
The paragraph has been revised and now reads:
"In bottom-up estimates, the process that produced the emission is modeled based on observations that constrain that process. For example, if the cause of the CO emissions is a wildfire, the emissions can be estimated based on knowledge about the burnt vegetation and the intensity of the fire. Conversely, in top-down estimates, the concentrations that resulted from the emissions are measured and traced back to their source. Again using the example of wildfire CO emissions, their effect in the atmosphere is an elevated CO concentration, that can be observed and then traced back and attributed to its source using atmospheric modeling."

*8) Line 42-45. 'direct observations of the source event', 'loose observational requirements'. Please rephrase to make it more accurate. also, 'potentially more elaborate' - why potentially?*
The paragraph has been revised. Regarding 'potentially': The term was used because some bottom-up models are very elaborate and some top-down approaches are rather simplistic. Therefore, it is not generally true that top-down approaches always require more elaborate assumptions than bottom-up approaches. The revised paragraph now reads:
"Both approaches are subject to various sources of error. Bottom-up estimates typically require direct observations of the source event (e.g., to have remote sensing information on fire intensity in the case of biomass burning) in addition to certain assumptions about the source itself, such as fuel characterization (ecosystem type, fuel loading, and fuel consumption rates) and emission factors in the case of biomass burning. Top-down estimates do not necessarily require observations of the source event itself, but rather observation(s) of the resulting concentrations at some point in the future are sufficient. However, while the observational requirements of top-down estimates are less strict, they often require a set of more elaborate assumptions for the atmospheric modeling, for example about chemistry and atmospheric transport."

*9) Line 48. 'top-down approach in the form of inverse modeling'. are there other forms?*
Some researchers distinguish data assimilation and inverse modeling as different categories of top-down approaches. To avoid confusion we have specified that this work is based on a 4DVAR approach and the sentence now reads:
"In this study, we use a top-down approach in the form of four-dimensional variational (4DVAR) inverse modeling, specifically, the state-of-the-art inverse modeling framework TM5-4DVAR."

*10) Line 58. 'including information from additional observations'. what do you mean by 'additional'?*
We meant 'additional' in the sense of 'not part of the prior already'. The text was updated to clarify:
"By incorporating information from additional observations beyond those used to create the a priori emissions, inverse modeling is able to reduce the uncertainties in the a priori emissions that are typically taken from bottom-up inventories."

*11) Line 83-96. The zooming capability is a very important point (strength) for this paper which should be highlighted more and taken advantage in extracting full information content of high-resolution datasets to address a science objective. Why would you reduce the observation resolution then? especially that this paper is considering: a) efficacy of high resolution obs, and b) near-real time application.*
In the current state of the model, zooming is limited to a resolution of 1°x1°, which is already much coarser than what TROPOMI can provide. As outlined above, the references to high-resolution observations + near real-time have been removed, as those were, indeed, not properly investigated in the presented study. The high-resolution observations provided by the TROPOMI instrument are still meaningful in this comparably low-resolution modeling study since the resulting super-observations will have a reduced uncertainty compared to each observation that went into them or to what a lower-resolution instrument could provide.

*12) Line 97. 'as a proof of concept'. has there been no inversions using tropomi yet?*
At the time of the initial manuscript submission, there were indeed no inversions using TROPOMI published yet. However, that has changed since, and throughout the manuscript, references to recent studies have been added, e.g. as outlined in the response to comment 3. above

*13) Line 117. Can you please elaborate why CMIP6 emissions are used?*
CMIP(6) emissions are widely used in the modeling community. They are adequate for inversions because they correctly predict where countries are and on what order of magnitude their emissions are. The inversion will correct for any potential mismatches between emissions and TROPOMI CO columns.

*14) Line 129-135. For inversions using full TROPOMI resolution, was the model resolution also change appropriately?*
No, the same zooming setup has been used in all experiments, to ensure comparability of the results. To avoid future confusion, the following sentence was added in L146, where the zooming setup is introduced:
"This zooming setup is used for all inversion experiments presented in this study."

*15) Line 137. Please elaborate on the rationale for the use of monthly OH from TransCom-CH4. Several studies have pointed out issues with using prescribed OH climatology especially for 'regional' inversions.*
OH in the global atmosphere is relatively well buffered, and the Spivakovsky climatology still complies with observed methyl chloroform loss rates. While we are aware of the limitations of the climatology, we are not aware of an alternative that is an objective improvement over the TransCom-OH. As pointed out in recent literature (e.g. Zhao et al. (2019) (`https://doi.org/10.5194/acp-19-13701-2019`) and Naus et al. (2019) (`https://doi.org/10.5194/acp-19-407-2019`)) there is a need for improved OH fields in the community, however, our current understanding and the currently available observations are insufficient to provide that at the moment. More appropriate OH fields are being explored in ongoing investigations. To address the raised concern, a small paragraph has been added to the manuscript (L153ff), which reads:
"Jiang et al. (2017) show that OH is well buffered in the atmosphere on a global scale over the past decades, as indicated by a low month-to-month variability in the methyl chloroform loss rate, and thus the TransCom OH climatology is still considered applicable to recent years, as in e.g. Naus et al. (2022)."

*16) Line 173. please elaborate on ' assimilating multiple datasets with different spatial and temporal resolutions at once and co-sampling of observations across datasets is neither necessary nor detrimental'. what do you mean by 'at once', 'co-sampling' and detrimental to?*
This statement was used to indicate that it is not necessary to only consider those observations across multiple datasets where they happen to occur at similar points in time and space and that it is not detrimental to the inversion results if they do. Since this constitutes unnecessary detail, it has been removed from the manuscript and the text now reads:
"Overall, in 4DVAR, the model is sampled temporally and spatially for each individual data point, and each point provides its own contribution to the cost function. As such, this approach is well suited to simultaneously assimilate multiple datasets with different spatial and temporal resolutions."

*17) Line 191. 'no daily cycles'. while it is consistent with OH, it is not suitable for inversions using 7km data. there's a mismatch in scale.*
Following Naus et al. (2022), who have shown a diurnal cycle in emissions to not be necessary even at 1°x1°, we neglected the daily cycle considering our inversion experiments are done at a resolution of at most 3°x2°, the observational data has at best a daily temporal resolution and the lifetime of CO is multiple weeks to months.

*18) Line 230-270. While the use of error correlation lengths is commendable, please elaborate on the scales use in the inversion? How are these calculated/estimated? This could be very useful to the community.*

As outlined in that section, the error correlation lengths were taken from previous studies with similar setups in the same model. We agree that having a rigorous approach to calculate them for a given setup would be very useful, however, that was not the focus of this study.

*19) Line 296-318. Please rephrase these paragraphs to make it clearer. It is unclear for example what is the exactly spinup period (and pinup inversion) and how was this conducted. For CO, this spinup period can be quite important, especially that there are differences in the model configuration of the initial conditions (incl. more importantly a different OH).*

The entire section "Initial conditions, spin-up, and main inversions" has been rearranged and in part rewritten and now reads:

[revised manuscript text omitted]

*20) Figure 2. Nice figure to better understand the elaborate gridding strategy. Still not sure though that this is appropriate for near real-time applications.*
The term near real-time has been removed from the manuscript, see response to comment 2. above.

*21) Line 517-524. Nice discussion on the limitation. It still begs the question (same as above comment) if this elaborate weighting strategy is appropriate.*
We agree that the described method will not be useful for near real-time applications as is, however, further developments on this basis are under investigation, some of which are aiming at significantly reducing the computational cost.

*22) Table 3. Very helpful to have.*
Thank you for the comment.

*23) The manuscript's honest discussion of inversion increments and identified issues is commendable, yet it risks obscuring key findings and blurring the clarity of its objectives. Reorganizing the structure and sharpening the focus would enhance clarity and impact.*
The manuscript has been reorganized as suggested to improve its structure and sharpen the focus.

---

## Author Comment (AC3)

We would like to thank reviewer 3 for the effort, interesting questions, and constructive comments. Below we address the points one by one. The reviewer's comments are listed in cursive, with our answers in blue and excerpts from the revised text in red.

*Review of Efficacy of high-resolution satellite observations in inverse modeling of carbon monoxide emissions using TM5-4dvar (r1258) by Johann Rasmus Nüß et al.*

*The article is quite comprehensive with many tests and evaluations. It is somehow frustrating that a lot of efforts has been put on looking at fire emissions with different priors and a setup attributing more errors to fire emissions, but in the end, the results not being presented (sect. 4.3.3). The paper would read better either with the inclusion of the fire emissions results, eventually with some explanation about why it does not work as intended to guide future studies, or by removing the sensitivity to prior fire emissions to make the paper more concise.*

We thank the reviewer for this comment. While we share the frustration, an in-depth analysis of the biomass burning emissions for the current setup is not meaningful, as pointed out in Sec 4.3.3. This focus on fire emissions was removed from the already lengthy manuscript for the sake of brevity at an early stage. TM5-4DVAR has been used for the analysis of biomass-burning events in the past, but some parts of the setup are now outdated. This paper introduces various improvements over earlier configurations, one of which is using the modern FINN2.5 as prior emissions. Since the GFED4.1s inventory is well-tested in TM5-4DVAR, the comparison is necessary to showcase the applicability of FINN2.5.

*The CO budget is not well presented with a lack of references to other CO inversions studies and other publications that compare inversion results, for instance Elguindi et al. (2020) Intercomparison of magnitudes and trends in anthropogenic surface emissions from bottom-up inventories, top-down estimates, and emission scenarios. Earth's Future, 8, e2020EF001520.* `https://doi.org/10.1029/2020EF001520`

We thank the reviewer for the valuable input and have extended the comparison to the budgets from other top-down CO emission studies, which can be found in L743ff of the revised manuscript and reads:

"Overall, the a posteriori secondary CO source is lower than the a priori production flux in all experiments, as can be seen in the global budgets provided in Table 3, where the posterior masses at the end of the inversion period (final masses) are consistently lower than the prior final masses. Naus et al. (2022), who used a similar setup, also found too high secondary CO production. All fluxes were extrapolated to annual budget terms in Tg CO yr$^{-1}$. This extrapolation may lead to misleading results when compared to budget terms published elsewhere, because the inversion period of the main inversions includes the biomass burning season, but excludes the increased anthropogenic emissions due to heating during most of the northern hemispheric winter. With this caveat in mind, we compare our prior and posterior budget terms with values from other inversion studies with different setups, namely to Jiang et al. (2017), who assimilated MOPITT CO and methyl chloroform surface measurements in the GEOS-Chem model, Müller et al. (2018), who assimilated IASI CO in the IMAGES model, and Zheng et al. (2019), who assimilated MOPITT CO in the LMDz-SACS model. A detailed comparison of these three studies can be found in Elguindi et al, (2020). Compared to the results of either of those studies, our extrapolated annual a priori budget terms for secondary CO production and chemical loss of CO to OH are far too large. However, our posterior chemical loss falls between the values found in Müller et al. (2018) and Zheng et al. (2019) and our posterior secondary CO production, while still larger, is much closer to what those studies found than our prior. This improved agreement implies that our a posteriori terms are more realistic than the a pirori ones. Note that our secondary production implicitly includes ocean and biogenic CO. While the total production and loss terms show reasonably good agreement with the aforementioned studies, the partitioning by source category of our emission terms differs slightly. Our anthropogenic/fossil fuel a posteriori CO is close to that found by Müller et al. (2018) and Jiang et al. (2017), but significantly lower than that reported by Zheng et al. (2019). In contrast, our biomass burning estimate is close to the multi-year mean of

Zheng et al. (2019). However, due to the high year-to-year variability in biomass burning emissions, as shown by both Müller et al. (2018) and Zheng et al. (2019), this result is difficult to interpret, especially since neither study covers 2018."

*Minor comments:*
*The title is misleading as most of the work is done with large scale setup for monthly emission inversions, and is focus on the comparison of TROPOMI inversions with global in-situ network.*
We agree with this assessment. The title has been revised to:
"Top-down CO emission estimates using TROPOMI CO data in the TM5-4DVAR (r1258) inverse modeling suit"

*Abstract: "Compared to the bottom-up estimates, all experiments result in strong (by up to 75%) broad-scale emission reductions in China and India. In part, the reduction over China can be attributed to policy changes." Explain the time period, the differences can be in absolute sense, or because of the change in emissions over time, please clarify.*
The abstract has been updated to reflect the requested clarification. The sentence now reads:
"Compared to the bottom-up estimates, all experiments result in strong (by up to 75 %) broad-scale emission reductions in China and India throughout the entire inversion period."

*L81: "spatial sampling of IASI (up to about 25×25km2; Clerbaux et al., 2009)." The citation indicates that IASI has footprints with diameters of 12 km diameter footprint. It is confusing because it looks like the MOPITT pixels are of similar size compared to IASI*
The spatial sampling of a satellite is not always the same as its footprint size. Unlike TROPOMI and MOPITT, where the spatial sampling and footprint size are the same, i.e. two adjacent footprints touch, for IASI the footprints are much smaller, than the distance between their center points. Every 50km view of IASI has four 12km footprints loosely arranged in a square (compare Fig 1 in Clerbaux et al., 2009). This leads to the distance between the center points of two adjacent footprints, the spatial sampling, to be around 25km. This quantity is more relevant for global scale inversions because it informs on the amount of data to be analyzed and the size of structures that can be resolved. The information on the footprint size has been added to the relevant places in the manuscript to reduce confusion, which now reads:
"Furthermore, TROPOMI procures CO observations at a high spatial resolution of up to $7 \times 7\,\mathrm{km}^2$ (Veefkind et al., 2012), which is roughly 10 times higher than the resolution of MOPITT (up to about $22 \times 22\,\mathrm{km}^2$; Drummond et al., 2010) and the spatial sampling of IASI (up to about $25 \times 25\,\mathrm{km}^2$ with 12 km diameter footprints; Clerbaux et al., 2009)."
and:
"For example, an empirically chosen variance inflation of 2 was used in Chevallier (2007) for Orbiting Carbon Observatory (OCO) $CO_2$ observations gridded to $3.75° \times 2.5°$, an inflation of 50 was used in Hooghiemstra et al. (2012a) and Naus et al. (2022) for MOPITT V4 (gridded to $1° \times 1°$) and V8 CO observations, respectively, and an inflation of again 50 was used in both Krol et al. (2013) and Nechita-Banda et al. (2018) for IASI CO observations at their native sampling resolution of up to about $25 \times 25\,\mathrm{km}^2$, with footprints of at least 12 km diameter."

*L142: 2.2 4DVAR approach: I am sorry if I missed it, but what is the assimilation window ?*
We may have used a different terminology here. The inversion periods (the time spans for which the model runs and data are considered) are 2018/01/01-2018/07/01 and 2018/06/01-2019/01/01 for the spin-up and main inversions, respectively (see Sec 2.3.3). The resolution of the state (the time step at which the model is allowed to scale the emissions in each category) is 1 month for both secondary CO production and fossil fuel CO and 1 day for biomass burning CO (see Sec 2.3.1).

*L227: "the a prior error is set to zero over the ocean" Chose between a priori or prior*
Fixed.

*L248: "Therefore, we use an exponentially decreasing correlation time of 9.5 months x for the secondary CO production at different times from the same cell." This seems to be a long time as CO itself has an average lifetime of 2 to 3 months.*

The correlation time denotes how long changes in the overall production patterns are expected to persist in time. This is independent of the lifetime of CO and more related to the lifetime of CO's precursor species (e.g. methane) and the time scales at which we expect the sources of those precursors to change.

*L265: "and the fairly up to date inventory (with historical data up to 2014 and projected data from 2015 onwards)," It is not really up to date. But what matters in the end is how the scenario matches the observations.*

For the time period investigated in this study, the year 2018, and at the time of initial creation of this study, CMIP6 was one of the most up-to-date global anthropogenic emission inventories. Alternatives, such as the CAMS-GLOBAL-ANT_v2.3 inventory, had the same limitation of providing only projected data after 2014. Considering that changes in anthropogenic emissions on a global scale are usually fairly predictable over relatively short time spans such as three years, the CMIP6 emissions should be adequate for inversions because they correctly predict where countries are and on what order of magnitude their emissions are. The inversion will correct for any potential mismatches between emissions and TROPOMI CO columns.

*L728: "Regardless, our extrapolated annual a posteriori budget terms are much closer to the ones found in literature (e.g. Zheng et al., 2019) than the a priori terms, implying that the a posteriori terms are more realistic." Please explain and clarify, maybe cite more paper about CO budgets, it is an interesting part of the paper that is not really complete at the moment.*

As mentioned in response to the second comment above, the comparison to the budgets from other top-down CO emission studies has been extended, and relevant literature has been added. Further investigations of the implications of this comparison are the subject of a future study.

---

## Author Response (AR2)

We thank the editor for the helpful comments on how to improve the clarity of our manuscript. Below we address the points raised by the editor one by one. The editors' comments are listed in *cursive*, with our answers in blue and excerpts from the revised text in red. Any line numbers refer to the latest manuscript version with mark-up.

*L51: You may like to omit using "Again" in this sentence, or rephrase "Using the same example ..."*
The sentence has been changed as advised and now reads (L45):
"Using the same example of wildfire CO emissions, their effect in the atmosphere is an elevated CO concentration that can be observed and then traced back and attributed to its source using atmospheric modeling."

*LL57–59: Please rephrase this sentence (grammar, the second part leaves ambiguity about meaning).*
To reduce ambiguity, the sentence has been split in two parts and the second part was further elaborated. It now reads (LL50–53):
"Top-down estimates do not necessarily require observations of the source event itself. Instead, it is usually sufficient to gather observations of the resulting atmospheric tracer concentrations during the time span between the source event and them falling below the detection limit due to loss processes and dispersion."

*LL167–170: Looking at Fig.4 of Jiang et al. (2017) I cannot confirm low (implied insignificant) month-to-month variation in MCF loss rate. Perhaps, you imply variation in annual averages? Please use accurately the statistical terms here (and throughout the manuscript), "variability" and "variation" are not the same (see, e.g., `https://web.ma.utexas.edu/users/mks/statmistakes/terminologyrevariability.html`). Whilst we do not know the (spatiotemporal) variability in the MCF rate not being able to measure it everywhere/every time in the atmosphere, we can analyse the variations in the limited set of estimates of average loss rate in time.*
We thank the reviewer for pointing out these mistakes. The word variation was replaced by the more accurate term variability, in places where it referred to a natural variation (L245, L285, L384, L715, and in the caption of Table 2). Additionally, we have changed the sentence in LL155–157 to more accurately cite Jiang et al. (2017):
"Jiang et al. (2017) show that OH is well buffered in the atmosphere on a global scale over the past decades, as indicated by the methyl chloroform loss rate varying by only 0.2 % between 2001 and 2015."

*L170: You probably refer to Naus et al. (2021) here, not Naus et al. (2022)? The former study analyses a set of available OH distributions. Reference: Naus, S., Montzka, S. A., Patra, P. K., and Krol, M. C.: A three-dimensional-model inversion of methyl chloroform to constrain the atmospheric oxidative capacity, Atmos. Chem. Phys., 21, 4809–4824, `https://doi.org/10.5194/acp-21-4809-2021`, 2021.*
The citation of Naus et al. (2022) was intentional. The sentence has been restructured to clarify and now reads (LL157–159):
"Thus, the TransCom OH climatology is still considered appropriate for studies investigating recent years. For example, Naus et al. (2022) use it in the context of inverse modeling of CO emissions up to and including the year 2018."

*L357: Please explicate on "in remote regions where transport is slow". What kind of transport is slow and why? I can only think of significant convective transport changes between land and ocean, however that would imply marine remote regions, for example.*
Here we conflated slow vertical transport and long transport times to remote regions. The ambiguity should be removed in the revised text, which now reads (LL329–332):
"Overall, harmonizing the mixing ratios modeled in TM5-4DVAR and the observations requires that the model is run over a longer period of time. Such a long spin up period is particularly relevant for high altitude layers, to which transport through vertical mixing is slow, or regions at large distances

from primary sources, to which transport takes a long time."

*L363: "Generated emissions" may be ambiguous for some readers (including me), are the emissions optimised during spin-up inversion implied?*
Yes, the optimized emissions are implied. The sentence was changed accordingly (L338):
"Therefore, the final month of the spin-up inversion is considered as its spin-down period, during which confidence in the optimized emissions and the resulting mixing ratios is reduced."

*LL779–799 and Table 3: I find double ambiguity in providing extrapolated budgets and comparing these with the results of the studies that do not even cover 2018, and fear that such extrapolated figures can be of little use for any follow-up study. I suggest recompiling Table 3 so that it includes only exact estimates (that is, Jun-Dec 2018), or both those of spin-up and main inversion periods (also exact). These, however, will be comparable by anyone who simulates entire 2018, including a possible extrapolation to their annual estimates. Should you like to keep the comparison on the extrapolated terms, please add a short explication on these are obtained from the exact terms, therefore you can omit quoting final extrapolated figures but keep the discussion.*
We thank the editor for pointing out this double ambiguity. First, in our initial draft we had intentionally only compared our results to Zheng et al. (2019) (who studied CO emissions up to 2017) because of the ambiguity introduced by comparing results from different time periods and we are not aware of global studies that specifically include 2018. We are aware that the other studies we added as requested by one of the reviewers are further removed in time. However, we would like to point out that the year-to-year variations within Jiang et al. (2017) and Zheng et al. (2019), respectively, are smaller than those between each of those studies and ours. As such, the systematic differences as discussed in manuscript still hold. Second, the magnitude of the biases introduced by comparing budgets for seven months with annual budget terms can be estimated by considering the impact this had on the prior emission, since they are known for the full year. Overall, these biases are small ($+4\%$ for biomass burning and secondary production; $< -2\%$ for fossil fuel) compared to the differences of our results to those reported in other studies. To reduce ambiguity we have removed the improper use of terminology of 'extrapolated annual budget terms', since simply using the unit 'Tg CO $yr^{-1}$' for a flux is not an extrapolation. The paragraph in question now reads (LL753–766):
"All fluxes in Table 3 are provided in Tg CO $yr^{-1}$, despite neither inversion period spanning a full year. While this unit allows for an easy comparison to (annual) budget terms published elsewhere, such a comparison must consider that the inversion period of the main inversions includes the biomass burning season, but excludes the increased anthropogenic emissions due to heating during part of the northern hemispheric winter. The biases of such a comparison can be estimated by comparing the prior fluxes from Table 3 for the *reference* inversion to the respective annual budgets of the prior source estimates, which show an overestimation by around $4\%$ for biomass burning (FINN2.5) and secondary CO production (from TM5-MP) and underestimation by less than $2\%$ for the anthropogenic emissions (CMIP6). With this caveat in mind, we compare our prior and posterior budget terms with values from other inversion studies with different setups, namely to Jiang et al. (2017, who assimilated MOPITT CO and methyl chloroform surface measurements in the GEOS-Chem model, Müller et al. (2018), who assimilated IASI CO in the IMAGES model, and Zheng et al. (2019), who assimilated MOPITT CO in the LMDz-SACS model. A detailed comparison of these three studies can be found in Elguindi et al. (2020). Compared to the results of either of those studies, our a priori budget terms for secondary CO production and chemical loss of CO to OH are far too large."
The caption of Table 3 was changed to:
"Note that the unit Tg CO $yr^{-1}$ for the columns showing rates was chosen for ease of comparison to other estimates and does not imply annual rates. The rates were obtained from the processed masses divided by the duration of the respective inversion periods, January to June (6 months) for the spin-up inversion and June to December (7 months) for the main inversions."

*Table 1: The value for inflation in 'Set 2 – Satellite only' entry is provided in parentheses. Do the*

*denote anything particular? If so, please include the respective description in the table caption or in a table footnote.*

The parentheses were meant to highlight that the inflation factor for the satellite only inversion was not calculated, but rather taken from the reference inversion, as stated in the caption. For clarity, this information was further explicated and moved to the table footnote, which now reads:

"†The inflation factor for the *satellite only* inversion cannot be derived as described in Sect. 3.2.2 since the flask measurements do not contribute to the observational cost in this experiment. Instead, the same inflation factor as for the *reference* inversion is used to ensure consistent weighting against the prior."